# Combining observations and simulations to investigate the small-scale variability of surface solar irradiance under continental cumulus clouds

Zili He[1], Quentin Libois[1], Najda Villefranque[1], Hartwig Deneke[2], Jonas Witthuhn[2], and Fleur Couvreux[1]

[1]CNRM, Université de Toulouse, Meteo-France, CNRS, Toulouse, France
[2]Department: Remote Sensing of Atmospheric Processes, Leibniz Institute for Tropospheric Research, Leipzig, Germany

**Correspondence:** Quentin Libois (quentin.libois@meteo.fr)

**Abstract.** The amount of solar radiation reaching the Earth surface (SSI) is critical for a variety of applications, ranging from surface-atmosphere interactions to solar energy. SSI is characterized by a large spatiotemporal variability, in particular in the presence of cumulus clouds. This results in complex spatial patterns of shadows and sunlight directly related to clouds' geometry and physical properties. Although key in many respects, the instantaneous spatial distribution of SSI remains largely unexplored. Here, we use unique observations from a dense network of pyranometers deployed during the HOPE field campaign to investigate the SSI spatial distribution. For cumulus scenes, bimodal distributions are found, with one mode corresponding to cloud shadows and the other to sunlit areas with enhanced SSI exceeding clear-sky values. Combining large-eddy simulations of cumulus clouds with Monte Carlo ray tracing, we demonstrate the capability of advanced numerical tools to reproduce the observed distributions and quantify the impact of cloud geometrical and physical properties on both modes. In particular, cloud cover strongly modulates their amplitudes, in addition to their position and width, which are also sensitive to cloud height, geometrical depth, and liquid water content. Combining observations and simulations, we also explore sampling strategies to estimate the SSI spatial distribution with a limited number of sensors, suggesting that 10 pyranometers integrated over 10 min can capture most details of the full distribution. Such a strategy could be used for future campaigns to further investigate SSI distributions and their impact on land-atmosphere exchanges or photovoltaic farm management.

## 1 Introduction

The amount of solar radiation reaching the Earth surface (hereafter referred to as SSI for surface solar irradiance) can be very variable in space and time, especially under broken cloud conditions (Long et al., 2006; Berg et al., 2011). In such conditions, SSI can even exceed clear-sky values when the sun remains visible in between clouds due to reflection by the cloud sides, a process often reported as cloud enhancement (Emck and Richter, 2008; Yordanov et al., 2012; de Andrade and Tiba, 2016). Although ubiquitous and very well known in the solar energy community (Lappalainen and Kleissl, 2020), this cloud enhancement has not been much investigated in the atmospheric science community, primarily because it is thought to vanish with spatial and temporal averaging on scales relevant to energetic transfers in the Earth system, even though recent work has demonstrated that systematic biases could remain even on daily averages (Gristey et al., 2020b). This phenomenon, and

more generally all radiative processes implying horizontal transfers in the presence of clouds, sometimes called 3D radiative effects of clouds, also remain overlooked in the atmospheric radiative transfer modelling community because most radiative transfer models embedded in atmospheric models rely on the plane parallel hypothesis, which inherently precludes to simulate such features (Várnai and Davies, 1999; Villefranque and Hogan, 2021). However, the spatial heterogeneity of SSI under broken cloud conditions is critical for the surface energy budget and land-atmosphere interactions (Lohou and Patton, 2014), the development of small-scale convection (Jakub and Mayer, 2017; Veerman et al., 2022), as well as for the stability of electrical systems fed by solar energy (Alam et al., 2014; Lohmann et al., 2016; Lohmann, 2018) or for urban thermal studies (Pacifici et al., 2019; Sanchez et al., 2021). As an illustration, the production of photovoltaic (PV) panels is very local, and the management of a PV plant is sensitive to small-scale irradiance variations because the time response of a PV system is nearly instantaneous (Gueymard, 2017). Although the complexity of the SSI spatial distribution is currently uncaptured by standard atmospheric models, the need from various sectors to better anticipate the detailed impact of clouds on SSI is now challenging the atmospheric science community.

While the instantaneous SSI spatial distribution is key for many applications, it remains difficult to assess. Standard SSI measurements are generally punctual and can only capture local temporal variations so that fast temporal variations are much more documented than small-scale spatial gradients (Inman et al., 2016). Satellite observations can provide a two-dimensional view of the Earth surface, but the spatial resolution of SSI satellite products is generally coarse compared to that of individual clouds, and estimating SSI from above requires many assumptions (Qu et al., 2017). Moreover, standard retrieval algorithms cannot capture cloud enhancement (Huang et al., 2019), making such products inadequate to investigate the details of the SSI spatial distribution (Beyer, 2016). As a result, most we know about SSI spatial variability comes from modelling. For decades, large-eddy simulations (LES) have been used to simulate cloud fields (Brown et al., 2002; Siebesma et al., 2003), and they now allow the simulation of extremely realistic clouds (Villefranque et al., 2019). These clouds have been extensively evaluated in terms of their geometrical and physical properties, often based on comparisons between spatial averages over the LES domain and vertical profiles observations (Neggers et al., 2003; Oue et al., 2016; Endo et al., 2019), but much less in terms of their radiative impact. Yet, assessing SSI fields would be a stringent test for the LES simulations, as such fields are sensitive to all geometrical and physical details of the simulated clouds. Only recently have a few studies carefully looked at SSI fields by combining LES with online or offline 3D radiative transfer models (Jakub and Mayer, 2017; Gristey et al., 2020b; Veerman et al., 2022). Gristey et al. (2020a) showed, for instance, that the features of the SSI spatial distribution under cumulus clouds are directly related to the macroscopic organization and physical properties of the clouds. This is promising to better characterize these clouds, which are particularly difficult to observe from space due to their small size. To avoid the computational burden of 3D radiative transfer simulations Tijhuis et al. (2023) proposed a method to reconstruct realistic SSI spatial distributions from plane parallel simulations. To this end, they applied a Gaussian filter to the SSI fields obtained under the plane-parallel hypothesis, allowing cloudy diffuse radiation to artificially spread over directly illuminated areas, somehow mimicking 3D effects. Yet, so far, the observational equivalent of such SSI spatial distributions is still largely missing, although a few field campaigns have already investigated related questions.

For instance, a network of 17 pyranometers was deployed around Kalaeloa airport on Oahu, Hawaii, from March 2010 to October 2011 (Sengupta and Andreas, 2010). This network has been used to investigate the power spectra of irradiance time series for individual sensors and for their average (Tabar et al., 2014) and to extract 2D fields of cloud motion vectors from ground-based observations of cloud shadows (Weigl et al., 2012). Luger et al. (2013) also used a grid of irradiance sensors to estimate the SSI spatial distribution on a PV farm and extract cloud velocity vectors. The HOPE field campaign (Macke et al., 2017), which took place in 2013 around Jülich, Germany, focused on the small-scale interactions between the surface and the atmosphere, in particular for the evaluation of subgrid processes in atmospheric models. During the campaign, an original instrumental system comprising 99 pyranometers was deployed for the first time (Madhavan et al., 2016). These observations have been carefully analyzed by Madhavan et al. (2017), with a main focus on the correlations between observations made by different sensors. These authors primarily aimed at quantifying the representativity of a single sensor for a neighbouring area. In particular, they showed that correlations arise at different spatial scales depending on the cloud regime. However, they did not focus much on the instantaneous SSI spatial distributions. Using the same dataset, Lohmann et al. (2016) focused on the correlations between time series to better predict local changes of SSI but did not look at the spatial distributions either. This dataset is, however, promising for investigating, from an observational point of view, the spatial variability of SSI. More recently, Mol et al. (2024) used a dense network of 20 to 25 radiometers to investigate the impact of clouds on SSI spatial patterns, focusing in particular on the spectral dimension of SSI. Other studies attempted to construct spatial fields of SSI, for instance, using a network of sky-imagers to locate clouds in the sky and then project their shadows at the surface (Nouri et al., 2022). However, in such cases, clouds are attributed an average transmissivity (Nouri et al., 2019) that does not capture the complexity of the radiation field in and around the cloud shadows. Kuhn et al. (2017) alternatively used a shadow camera to estimate SSI fields with an accuracy of about 10 % but did not discuss how the measurement errors modified the overall distribution.

With the existing literature on SSI spatial variability in mind, the main objective of the present study is to investigate instantaneous SSI spatial distributions under broken cloud conditions by combining the unique observations from the HOPE dataset with simulated SSI fields obtained by running 3D radiative transfer on LES simulated clouds. In line with previous studies addressing this question we focus on cumulus clouds because they are responsible for the largest small-scale variability of SSI. These clouds, ubiquitous across a large fraction of the globe, also remain a challenge for weather and climate modelling, primarily because their small size means that they are generally parameterized, and their radiative impact as well. To identify situations from the HOPE dataset corresponding to golden cases of cumulus clouds, i.e., very homogeneous fields close to those simulated by ideal LES, we propose an original selection strategy. The comparison of these golden cases to simulations suggests that simulations are appropriate for studying SSI spatial distributions. Hence, building on this first general assessment of instantaneous SSI spatial distributions, we then tackle two independent questions. We first explore measurement strategies to capture the SSI spatial distribution with a limited network of radiation sensors, which is addressed by combining the observations and simulations. We then investigate how cloud properties control SSI spatial distributions, which is carried out by perturbing the cloud properties in the simulation system and quantifying the impact on SSI distributions.

Section 2 introduces the HOPE dataset, the LES model and the simulations, as well as the 3D Monte Carlo radiative transfer code. The methodology followed to answer the objectives of this study is then detailed in Sect. 3. The analysis of the SSI spatial distributions from both the observations and reference simulations are presented in Sect. 4, while Sect. 5 further investigates how instantaneous SSI spatial distributions can be approached by appropriate spatiotemporal sampling of SSI. Finally, sensitivity tests are performed in Sect. 6 to investigate the impact of cloud properties on the SSI fields. Section 7 summarizes the main results and gives some perspectives.

## 2  Data

### 2.1  Observations from the HOPE field campaign

The High Definition Clouds and Precipitation for advancing Climate Prediction (HD(CP)2) Observational Prototype Experiment (HOPE) campaign (Macke et al., 2017) was designed to evaluate the German community atmospheric model (ICON) and to learn about atmospheric physics at spatiotemporal scales at which processes are parameterized in the model. To this end, observations of aerosols, clouds, and precipitation were collected with high spatial and temporal resolutions near Jülich, Germany (50.909°N, 6.4139°E, 111 m asl) in April and May 2013.

During this campaign, and until July 2013, a high-density network of 99 pyranometers was deployed (although some of them have not been working all the time) on a $10 \times 12$ km$^2$ area (Madhavan et al., 2017). The minimum and maximum distances between any two pyranometers are 0.14 and 14.1 km, and the mean distance to the closest neighbour is 0.86 km. The SSI (i.e. the downwelling solar flux density per unit of horizontal surface in W m$^{-2}$) was measured at 10 Hz and then averaged at 1 Hz. The low-cost silicon sensors used are only sensitive across the spectral range of 300-1100 nm, hence SSI retrieval requires a calibration step. As the spectral distribution of SSI varies depending on atmospheric conditions (Lindsay et al., 2020), this calibration can result in errors up to 5%.

In addition to these pyranometers, the Leipzig Aerosol and Cloud Remote Observations System (LACROS) station was deployed at Krauthausen (50.880°N, 6.415°E, 99 m asl) in April and May, the two-month period on which we focus in this study. The station includes a 35-GHz cloud radar and a lidar ceilometer, from which cloud boundaries (cloud base and top heights) can be retrieved, a microwave radiometer measuring liquid water path (LWP, in kg m$^{-2}$), as well as an all-sky imager (Bühl et al., 2013).

Figure 1 shows a sample of this dataset on a day with fair-weather cumulus clouds, as can be seen on the image captured by the all-sky imager (Fig. 1(a)). Figure 1(b) shows the instantaneous SSI measured by the pyranometer network. Small SSI values around 500 W m$^{-2}$ (blue points) correspond to cloud shadows, while large values closer to 1000 W m$^{-2}$ (red points) correspond to clear sky. It can be noticed that clear-sky values are heterogeneous in space, a point that will be specifically addressed later.

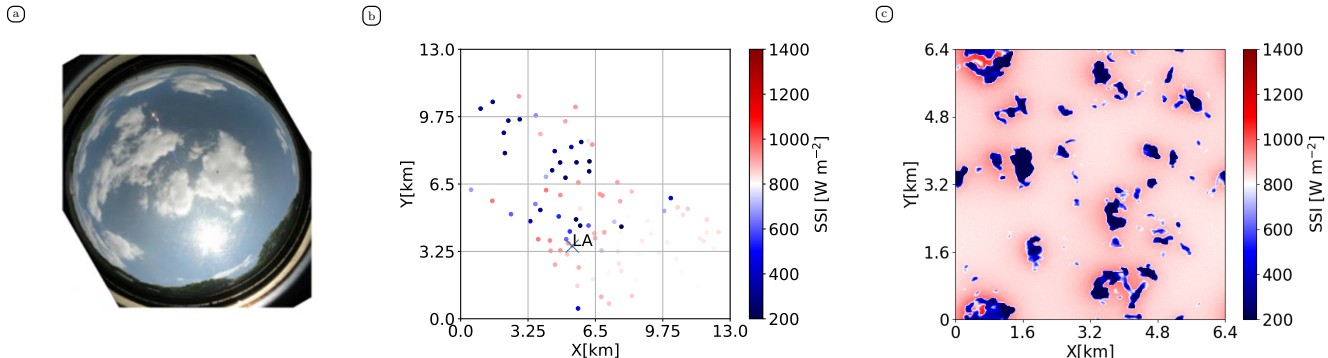

**Figure 1.** Illustrations of data used in this study. (a) All-sky image at 10:37:30 UTC on 5 May 2013. The picture was taken at LACROS station, which is marked as "LA" in (b). (b) Mean SSI over one minute, measured by the 99 pyranometers from 10:37 to 10:38 UTC on 5 May 2013. (c) Simulated SSI under a synthetic cumulus cloud field simulated by a Meso-NH LES under the same solar zenith angle as in (b).

## 2.2 Large-eddy simulations

To complement the HOPE observations, a high-resolution simulation of a golden case of developing cumuli over land, namely the ARM cumulus case (Brown et al., 2002), is used. The simulation is run with the Meso-NH model (Lac et al., 2018) for 15 hours over a 6.4 km wide (domain size similar to the area covered by the pyranometer network) and 4 km high periodic domain with horizontal and vertical resolutions of 25 m. Three-dimensional fields of liquid water content (LWC), specific humidity, temperature, pressure, and wind are output every minute during the cloudy hours of the simulation. The model uses an anelastic system of equations and a 3D turbulent kinetic energy scheme (Cuxart et al., 2000) with a diagnostic mixing length (Deardorff, 1980). For the advection of meteorological and scalar variables, discretization of the spatial derivative is based on a piecewise parabolic method, enabling the scheme to handle sharp gradients and discontinuities very accurately. Time integration is forward-in-time. Advection of momentum is solved using a centred discretization of the fourth order in space and a Runge-Kutta-centered fourth-order scheme in time. The water phase transformations are parameterized with the ICE3 one-moment microphysical scheme (Pinty and Jabouille, 1998). Diurnally varying surface turbulent fluxes are prescribed during the simulation, as well as cooling and drying tendencies summarizing large-scale advection and radiative tendencies, as described in Brown et al. (2002).

## 2.3 Radiative transfer simulations

A 3D radiative transfer model based on Monte Carlo methods (Villefranque et al., 2019) is used to simulate SSI fields every minute of the fifth hour of the LES (10:30-11:30 LT) in offline mode. It uses the solar zenith angles (SZA) at Jülich on 5 May 2013, from 11:36 to 12:36 UTC, which decreases from 36.8 to 34.5°. Each pixel of each field is a $5 \times 5 \ m^2$ square. Note that a finer resolution than the LES is used to accurately simulate what happens near cloud shadow edges, where variations occur at smaller scales than the cloud resolution. Such a fine resolution allows to correctly simulate the rapid transition from the

shadow to the clear-sky areas and to capture the value of the maximum cloud enhancement, which is essential to reproduce the SSI distribution. Each pixel corresponds to an SSI estimate, calculated as the mean flux over 15000 photon-path realizations, resulting in a statistical uncertainty of approximately 1%. Following the $k$-distribution model, a quadrature point within the spectrum integral is sampled for each photon path, following the method proposed by Villefranque et al. (2019). This strategy is proven to be unbiased and has good convergence performance. An example of such a field is presented in Fig. 1(c). Three-dimensional fields of LWC and water vapour are used to compute the single scattering properties in the LES domain, which is periodically repeated on the horizontal. Importantly, the simulations are performed without aerosols, although they can significantly alter the SSI distribution (Gristey et al., 2022). The standard mid-latitude summer atmospheric profile, also used in the I3RC cumulus case (Cahalan et al., 2005), is used as background atmosphere above the domain. Gaseous absorption properties are computed for this background atmosphere using the correlated-$k$ model implemented in RRTMG (Iacono et al., 2008) and for twenty profiles with perturbed absorption coefficients. These pre-tabulated absorption coefficients are then interpolated to account for the actually simulated water vapour concentrations in the LES domain (see Appendix C.2.1 of Villefranque et al., 2019). Cloud droplets have a constant effective radius of 10 $\mu$m and an effective variance of 0.010. Their optical properties are computed from Mie calculations using the code developed by Mishchenko et al. (2002) and assuming a log-normal size distribution. The surface is assumed Lambertian with the spectral albedo of grass (Meso-Star, 2021). To simulate broadband solar fluxes, spectral integration is then performed from 0.3 to 4 $\mu$m.

## 3 Methods

### 3.1 Objective selection of cumulus cloud periods

As this study focuses on cumulus clouds, cumulus scenes need to be identified from the observations. In order to select one-hour-long periods when cumulus are present, four metrics are defined:

$$c_1(t) = \langle\langle \mathrm{CSI}(x,t)\rangle_x\rangle_t, \tag{1}$$

$$c_2(t) = \langle\sigma_x\left[\mathrm{CSI}(x,t)\right]\rangle_t, \tag{2}$$

$$c_3(t) = \sigma_t\left[\langle \mathrm{CSI}(x,t)\rangle_x\right], \tag{3}$$

$$c_4(t) = \sigma_x\left[\langle \mathrm{CSI}(x,t)\rangle_t\right], \tag{4}$$

where $\mathrm{CSI}(x,t)$ is the clear-sky index at location $x$ and time $t$, defined as $\mathrm{CSI} = \mathrm{SSI}/\mathrm{SSI}_{\mathrm{cs}}$ (Lohmann and Monahan, 2018), with $\mathrm{SSI}_{\mathrm{cs}}$ the theoretical clear-sky SSI estimated from a clear-sky model (Ineichen, 2008, 2016) embedded in the *pvlib* python package (Holmgren et al., 2018). This model accounts for climatological concentrations of aerosols, ozone, and water vapour. Yet, local conditions at the moment of the measurement might significantly differ from their climatologies, hence the CSI might be biased. However, this should not affect the identification of the cumulus cloud periods. $\langle.\rangle_u$ and $\sigma_u[.]$ denote average and standard deviation, respectively, taken over dimension $u$. Spatial dimension ($u = x$) implies that data are taken over all pyranometers, and temporal dimension ($u = t$) implies that data are taken over one hour centered on $t$.

$c_1$ and $c_2$ thus quantify the temporal average of the spatial mean and spatial variability of CSI, respectively. Hence large values of $c_1$ indicate situations where either no clouds are present, or they are present with a minor effect on radiation on average. It means that either their fractional cover or their optical depth is small. Combined with large values of $c_2$, which indicate high spatial variability of CSI, broken cloud situations can be detected and clear-sky or homogeneous optically thin clouds eliminated. $c_3$ quantifies the temporal variability of the averaged SSI over the domain, which allows the identification of stationary situations. Eventually, $c_4$ quantifies the spatial variability of the averaged SSI over the time period, which allows the selection of statistically uniform cloudy situations over the domain. Figure 2(a)-(b) shows the time series of the four metrics for selected periods in April and May.

To identify periods with broken cloud conditions, we follow a two-step process:

- Pre-selection based on $c_1$ and $c_2$: from our data, we identify periods where $c_1$ and $c_2$ are among the highest. Specifically, we look for the periods that fall within the top 30% for both $c_1$ and $c_2$ to focus on times when broken clouds are present. The selected periods are highlighted by red dots in Fig. 2.

- Complementary selection using $c_3$ and $c_4$: among these pre-selected periods, we apply another filter based on two additional criteria, such that $c_3$ and $c_4$ values are both among the lowest 30% of the pre-selected cases. This step helps refine the selection to periods where cumulus fields are temporally stable and spatially uniform.

After these two steps, five cases are identified and highlighted by red vertical lines in Fig. 2(a)-(b), on April 18[th], April 20[th], April 25[th] and May 5[th] (2 periods). Thanks to the all-sky images and to MODIS satellite images, it was verified that they indeed correspond to cumulus cloud situations, thereby validating our automatic selection procedure.

Figure 2(c) zooms on the selected periods and shows as well the metrics computed from the LES cumulus cloud field (where clear-sky SSI is estimated from a clear-sky radiative transfer simulation using the same Monte Carlo code and setting LWC to zero). The metrics $c_1$, $c_2$ and $c_3$ are very similar between the observations and the simulation. On the contrary, $c_4$ is significantly smaller in the simulation than in the observations, which suggests that the real situations still feature more spatial heterogeneity than the ideal case characterized by a uniform surface and periodic boundary conditions.

Figure 2(d) shows the values of the four metrics for several days identified as "broken clouds" by (Madhavan et al., 2017). The only intersection between their set of broken-cloud cases and ours is April 25th. Note, however, that Madhavan et al. (2017) selected entire days, whereas we selected only hours. Cases we selected might occur during days otherwise clear or overcast, hence not considered as broken-cloud days. It is puzzling, though, that we did not select more cases on days flagged as cumulus days by Madhavan et al. (2017). Looking at our metrics during these days, one can see that they are indeed characterized by high $c_2$ values, suggesting a spatially heterogeneous SSI, but considerably lower $c_1$ values than in our selected cases, suggesting larger cloud covers or optically thicker clouds. Furthermore, they are associated with larger $c_3$ and $c_4$ values, suggesting the periods are less temporally and spatially stable compared to the periods we identified in this work. This might indicate that our criteria are too restrictive. Note that actually, our method was designed to identify cumulus periods, but not necessarily all of them. In particular, the condition on $c_1$ could have been less strict. It is also possible that normalized standard deviation (divided by mean values) would have been more adapted than absolute standard deviations for $c_3$ and $c_4$. The selection procedure could

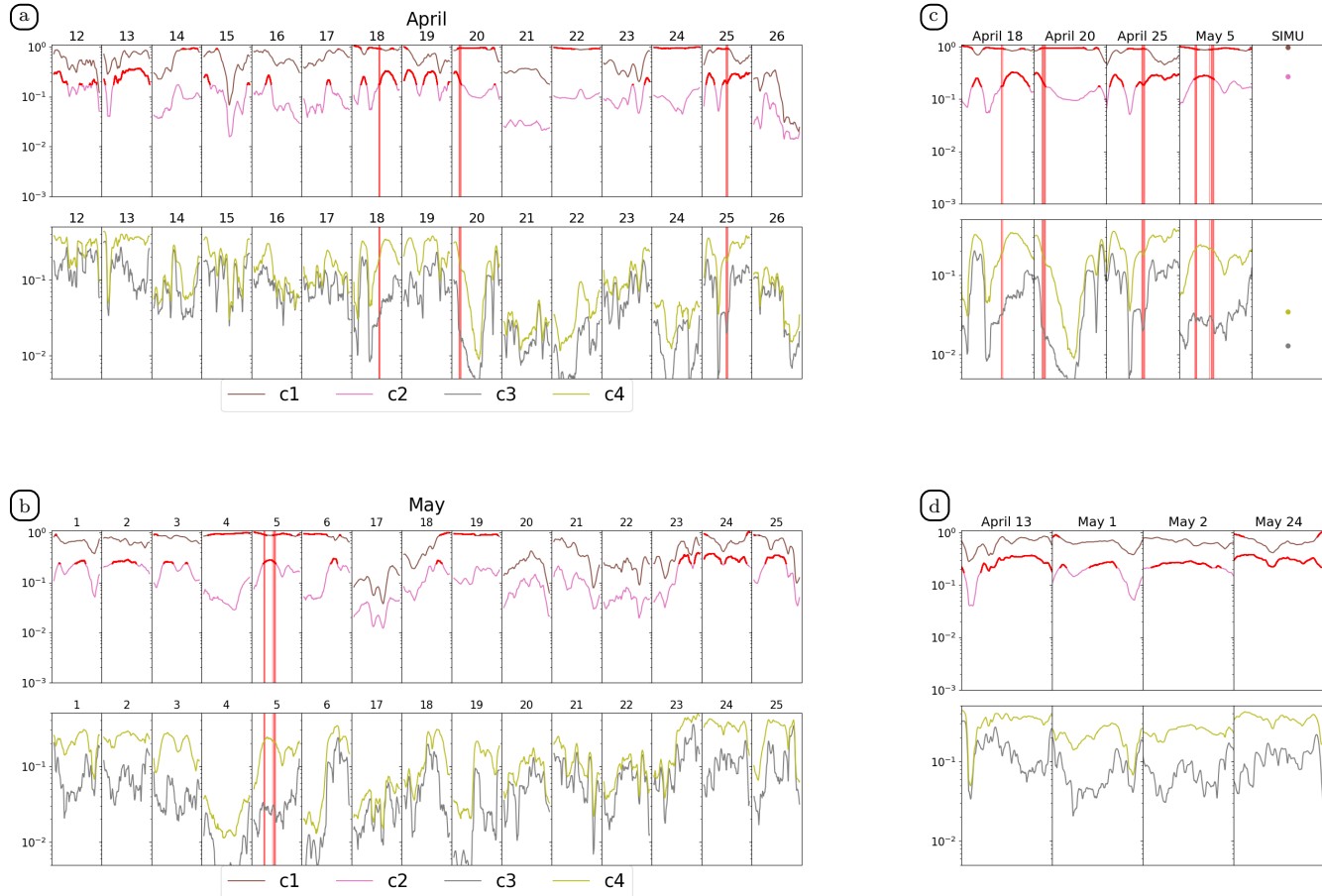

**Figure 2.** Time series of the four metrics $c_i$ in (a) April and (b) May. Values of the four metrics for (c) the periods identified as cumulus cloud hours following our method (highlighted by red vertical lines) and the simulation, and (d) for the periods identified as cumulus cloud days by Madhavan et al. (2017).

thus certainly be refined for future studies. However, for the present study, we will use the 5 selected cases highlighted in Fig. 2(c).

### 3.2 SSI distributions

To characterize the SSI spatial distribution, probability density functions (pdfs) of normalized SSI are used, in line with Gristey et al. (2020b). The normalization factor is simply the cosine of SZA at the time and location of measurement in the case of observations or of prescribed SZA in radiative transfer simulations in the case of LES data (in which case SZA values correspond to those of P5). Hereafter these pdfs are simply referred to as SSI distributions. Except when stated otherwise,

the SSI distributions are constructed by cumulating data at one-minute resolution during one hour (which implies 1-minute averages for the observations) and over the whole domain (99 pyranometers in the observational dataset, 1280 × 1280 grid points in the LES data). Bins are 30 W m$^{-2}$ wide.

The distributions are bimodal, with one mode corresponding to cloud shadows and the other to clear sky (inter-shadows gaps). It can be seen in Fig. 1(b)-(c) that the largest values correspond to clear-sky regions near cloud shadows being over-illuminated. Indeed, these regions receive additional radiation reflected by cloud sides, a 3D effect sometimes called enhancement, side leakage, or channelling, and well documented in the literature (e.g. Marshak and Davis, 2005).

Gristey et al. (2020a) used a neural network trained on LES data to show that the parameters of analytical functions matching each mode of the SSI distribution could be predicted from a few properties describing the cloud field. This implies that these distributions contain valuable information on the overlying cloud field. In the following, the distributions are characterized by the mean and standard deviation of subsamples corresponding to each mode without assuming particular distribution shapes. This is a way to condense the information and facilitate its interpretation. The cloud-shadow mode corresponds to values smaller than 500 W m$^{-2}$, whereas the clear-sky mode corresponds to values larger than 900 W m$^{-2}$. Values in between correspond to shadow edges. They are associated with low relative occurrence and are excluded from the systematic analysis. Although these two thresholds are arbitrary, the main objective was to qualitatively isolate both modes, which proved to be acceptable for the cases encountered. However, defining these modes in a more flexible way, which would depend on the actual distribution and would work for a larger variety of cloud properties, would be useful and should be considered for future studies. To compare two distributions obtained from different cloud fields or different datasets, root mean square deviations (RMSD) will be computed on the whole histogram and on each mode separately.

### 3.3 Modification of LES fields

Sensitivity tests are performed in Sect. 6 in order to gain physical insight on how various cloud characteristics drive SSI distributions. For each category of test, the 60 LES cloud fields of the one-hour-long simulation are modified, varying a single property at a time, among cloud LWC, cloud base height, cloud depth, or cloud fraction. The various categories of tests are:

- `LWCx`, where LWC in the clouds is uniformly scaled by a given factor (e.g., 0.6 or 1.4);

- $\Delta$H, where the full cloud layer is translated on the vertical (e.g., 400 m closer to the surface ($\Delta$ H =-400) or 400 m higher ($\Delta$ H =400));

- $\Delta$D, where cloud layer depth (D) is increased. First, each cloudy column is shifted upwards by $n$ layers of thickness $\Delta z$ (in the following, $n = 16$ and $\Delta z = 25$ m); that is, clouds are moved upwards by a distance $n\Delta z$. Then, $n$ layers below the new cloud base are filled with the same LWC as the original cloud-base layer. Finally, LWC is scaled column-wise so that the LWP field is unchanged: the whole field contains the same total mass of liquid water as the original one, but the maximum LWC is smaller;

**Table 1.** Cloud properties resulting from the sensitivity tests. Each property is given at the first/last timestep of the one-hour-long period. Symbol — indicates the same value as control.

| Case (units) | cloud base height (m) | cloud layer depth (m) | liquid water path (g m$^{-2}$) | max water content (mg kg$^{-1}$) | cloud cover (%) |
|---|---|---|---|---|---|
| Control | 825 / 875 | 350 / 700 | 1.6 / 9.4 | 9.7 / 34.0 | 14.3 / 27.7 |
| LWCx0.6 | — | — | 0.9 / 5.6 | 5.8 / 20.4 | — |
| LWCx1.4 | — | — | 2.2 / 13.1 | 13.6 / 47.5 | — |
| $\Delta$H=-400 | 425 / 475 | — | — | — | — |
| $\Delta$H=400 | 1225 / 1275 | — | — | — | — |
| $\Delta$D=400 | — | 750 / 1100 | — | 6.0 / 26.7 | — |
| CC=125 | — | — | 6.7 / 30.8 | 41.4 / 105.3 | 46.2 / 59.8 |

– CC, where cloud fraction at each layer (and thereby the total cloud cover (CC) seen from above as well), is increased. To this end a collection of translated cloud fields is first created by incrementally shifting the original cloud field in each horizontal direction (including diagonals), up to a given distance (e.g. 125 m). Then, the resulting translated cloud fields are averaged together. Finally, the resulting 3D field of LWC is uniformly scaled at each vertical level so that the original "in-cloud" LWC (defined in each model layer as horizontal domain average content divided by cloud fraction),
is unchanged.

In sensitivity tests LWCx and $\Delta$H, the impacts of changing LWC and cloud base height are well isolated. In $\Delta$D and CC, however, not only is the cloud geometry modified, but also the LWC distribution inside clouds and across the domain. In $\Delta$D, LWP is preserved, but the shape and absolute values of LWC vertical profiles are modified. This might result in unrealistic features in clouds, hence in SSI fields. In CC, layer-wise mean in-cloud LWC is preserved, but as cloud fraction increases,
the total amount of water in the domain also increases. Table 1 summarizes the various tests performed and the corresponding modifications of the cloud field.

## 4 SSI distributions in observations and simulations

Five cumulus periods of 1 hour were selected in the observations following the method detailed in Sect. 3.1. Figure 3(a) shows the SSI distributions for each period. For each case, an effective cloud cover is diagnosed by computing the fraction of the
260 measurements (99 pyranometers during one hour), with normalized SSI lower than 900 W m$^{-2}$. This is referred to as "shadow cover" in opposition to the "cloud cover" classically defined as the fraction of a domain covered by clouds when seen from above. Figure 3(a) also shows an SSI pdf for a clear-sky period, taken on 4 May 2013, 12:12-12:13 UTC, which was also flagged as clear sky by Madhavan et al. (2017) and confirmed by all-sky images.

Looking at SSI distributions in the presence of broken clouds, one can see that all cases are characterized by similar bimodal
distributions. Their properties differ between the various observed cases, although one interesting common feature, already

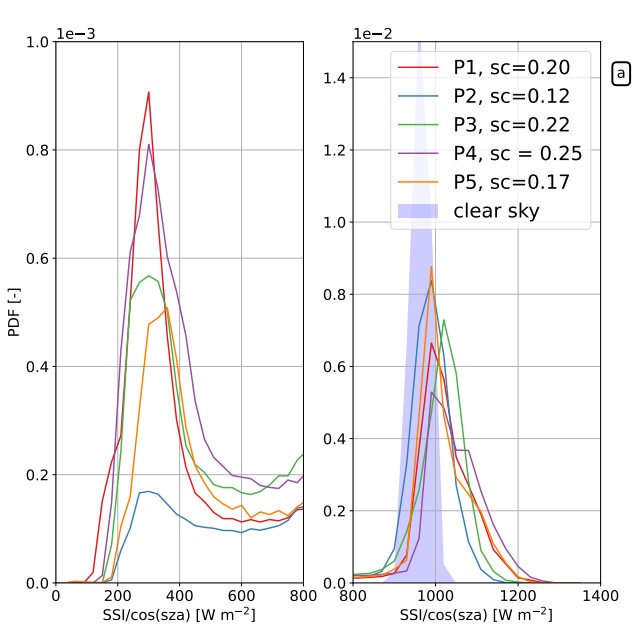
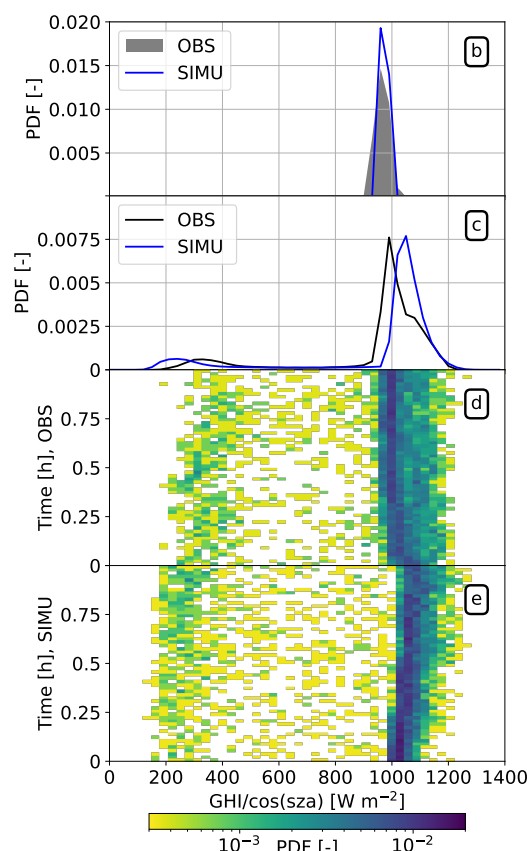

**Figure 3.** Observed and simulated normalized SSI distributions. (a) Distributions for the 5 selected periods in UTC: P1: 18 April 2013, 13:00-14:00; P2: 20 April 2013, 9:12-10:12; P3: 25 April 2013, 12:32-13:32; P4: 5 May 2013, 9:30-10:30; P5: 5 May 2013, 11:36-12:36. sc in the legend indicates the "shadow cover", which corresponds to the fraction of the surface occupied by cloud shadows (integration of the distribution from 0 to 900 W m$^{-2}$). The pdf for a clear-sky period (4 May 2013, 12:12-12:13 UTC) is also shown. (b) Distributions of observed (same as panel a) and simulated SSI during a clear-sky period of 1 minute. (c) Distributions of observed (P5) and simulated SSI Cumulated over the full hour. (d) Observed (P5) and (e) simulated distributions of the SSI as a function of time along the one-hour period.

pointed out by Gristey et al. (2020b), is that a large number of values are significantly larger than the values expected in clear-sky conditions — a typical signature of 3D radiative effects. In a sense, clouds act like the mirrors that are used to collect solar radiation in concentrated solar power systems. Interestingly, based on the shadow cover values, it seems that maximum cloud enhancement tends to increase with cloud cover. We believe that it is because the clear-sky region is receiving scattered radiation from more surrounding clouds (the sensitivity of the SSI distribution to cloud cover is investigated in Sect. 6).

Figure 3(b) presents simulated and observed distributions under clear-sky conditions. They are both unimodal and symmetric, with approximately the same width and around the same mean value, suggesting that the impact of aerosols, which are not accounted for in the simulations, was rather limited for that particular day. However, it is important to note that their widths have distinct origins. The observed distribution results from instrumental, intrinsic variability, as well as heterogeneity in atmospheric (e.g. water vapour, aerosols) and surface (e.g. albedo) properties, whereas variability in the simulations is dominated by Monte Carlo statistical noise. Both simulations and observations account for water vapour heterogeneity, but its effect on solar radiation is too small to explain the obtained standard deviations. Increasing the number of photons in the Monte Carlo simulation leads to a narrower distribution (not shown). Hence it is a coincidence that both sources of noise have the same amplitude here: the inherent lack of heterogeneity in the LES is somehow balanced to the right amount by Monte Carlo noise. Because these noises introduce much less variability than that caused by the presence and characteristics of clouds, in the remaining parts of this study, both simulations and observations are analyzed without further consideration of noise. Nevertheless, the detailed understanding of the observed clear-sky pdfs certainly deserves more attention to disentangle the sensors' inter-calibration issues from the actual spatial variability of SSI across the observed domain.

Figure 3(c) compares the observed (P5) and simulated SSI distributions cumulated over the full hour. As for P5, only 95 are used (four were not working during this specific period), and only 95 pixels were randomly sampled in the simulation for fair comparison. The distributions have very similar shapes, although the observed one shows a bump in the right part of the clear-sky mode that is not present in the simulated one. The cloud-shadow mode is also shifted towards lower values in the simulation compared to the observations, while the simulated clear-sky mode peaks at a greater normalized SSI than the observed one.

To further understand the cumulated distributions, Figures 3(d) and e show the SSI distributions for each minute of P5 and of the simulation. It is clear that the shape of the distributions is relatively constant throughout the hour. It can be seen, however, that the cloud cover (integral of the cloud-shadow mode) increases along the simulation, as already noticed in Table 1. This seems to increase 3D effects and, therefore, amplify the enhancement of SSI in clear-sky regions between cloud shadows, as suggested by the shift towards larger values of the clear-sky peak. In the observations, the clear-sky mode appears quite stationary, apart from the very beginning, which features larger values and probably explains the bump of the cumulated distribution (Fig. 3(c)). On the contrary, the cloud-shadow mode shifts to larger values with time, which could suggest that clouds are getting optically thinner or that light entrapment between the surface and the clouds is getting more intense (Hogan et al., 2019; Villefranque et al., 2023). It can also be noticed that small SSI values (that is, inside cloud shadows) are consistently smaller in the simulation than in observations, meaning that the observed and simulated cloud fields are probably distinct in terms of detailed cloud physical properties.

A detailed investigation would be needed to further understand all these differences, in particular, to disentangle the role of assumptions made in the LES and radiation code (idealized surface, limited area domain with periodic boundary conditions, approximate scattering phase function for cloud droplets, arbitrary and homogeneous value of the cloud droplet effective radius...) from the role of actual differences of cloud properties and geometry. This is out of scope here, as we only aimed to demonstrate that the combination of LES and Monte Carlo numerical tools is well suited to simulate realistic SSI distributions. To have a better match between observations and simulations, simulations should correspond to the same atmospheric and surface conditions as the observations, which is not the case here. Note that the LASSO experiment on the Atmospheric Radiation Measurements Southern Great Plain site was specially designed to allow strict comparison between observations and LES (Gustafson Jr et al., 2020) and would provide a very relevant framework to investigate these questions.

## 5 Sensitivity of SSI distributions to spatiotemporal sampling

We have shown in the previous section that cumulating SSI measurements from a dense network of 99 (or 95 in P5) pyranometers over one hour allows to capture most of the spatial variability of SSI. However, such instrumental configuration is unique to the HOPE campaign and cannot be practically deployed in all field campaigns. Hence this section aims at providing guidance on the measurement strategy needed to estimate instantaneous SSI distributions in the presence of broken clouds. To this end, we analyze the sensitivity of the distributions to the number of pyranometers used to compute this distribution and to the time period on which observations are cumulated. We apply the same strategy to the observations and simulations.

To gain insight into the way temporal and spatial sampling together operate, we first focus on the hourly distributions. Based on the P5 observations, we assess the deterioration of the full distribution (95 pyranometers over one hour) induced by either using fewer pyranometers or spanning a shorter period. Figure 4(a) shows the evolution of the RMSD between the approximate and the full distributions for various numbers of pyranometers and periods of integration (all symmetrical around the middle of the full period). For each period of integration, the subsampling is repeated for 512 different random combinations of the same number of pyranometers to characterize the uncertainty of the results. As expected, decreasing the number of measurement sites or the duration of integration increases the RMSD. The sensitivity to the time period seems quite independent of the number of pyranometers, as suggested by the fact that the curves are almost parallel to each other in Fig. 4(a). The individual contributions of the cloud-shadow and clear-sky modes are shown in Fig. 4(b), with a dominant contribution from the clear sky (expected from the larger values). When setting the integration period to 10 min, a time scale at which the cloud field can be considered stationary, at least 50 pyranometers over the $10 \times 12$ km$^2$ are needed to reduce the RMSD down below $5 \times 10^{-4}$, which is the minimum value needed to distinguish distributions corresponding to distinct cloud geometrical characteristics (see Sect. 6). In particular, we note a strong reduction of the clear-sky RMSD from $4.5 \times 10^{-3}$ to $5 \times 10^{-4}$. Examples of reconstructed distributions are shown in Fig. 4(c) and can be compared to the reference distribution represented by the pink shading. It shows that the sensitivity of the reconstructed distributions to integration time is smaller than their sensitivity to the number of sites and that 10 sites are essentially enough to capture the variability measured at all sites. A similar RMSD is

obtained when cumulating 5 pyranometers over 1 hour or 10 pyranometers over 10 min, or between 10 pyranometers over 1 hour and 50 pyranometers instantaneously.

The same analysis is now carried out for the simulation, where a larger number of measurement sites can be sampled (Fig. 4(d)). Here, the reference distribution to compute the RMSD contains all $1280 \times 1280$ pixels of the simulation, cumulated over one hour. Qualitatively, the sensitivity of RMSD to subsampling is similar to the observations, that is, the error increases when either the number of measurement sites or the integration time is reduced. However, the decrease of RMSD with integration time is much faster at shorter integration times in the simulation. This may be related to a stronger background

wind in the simulation ($10 \text{ m s}^{-1}$) than in P5 ($5 \text{ m s}^{-1}$ at 1 km altitude as measured by radiosoundings). There is an inflexion point around 600 s for the sensitivity to the integration time when a single measurement site is used (the time position of this inflexion point decreases with the number of sites). According to the mean wind in the simulation and assuming that clouds do not significantly evolve, this corresponds to a 5 km distance sampling. These tests suggest that the hourly distribution is captured satisfactorily (RMSD below $5 \times 10^{-4}$) when using 50 pyranometers over 4 min or 10 pyranometers over 20 min.

The decomposition of the RMSD between the clear-sky and cloud-shadow modes is shown in Fig. 4(e). Interestingly, clear-sky RMSD dominates for integration over less than 1000 s, while for larger integration times, both modes equally contribute. Figure 4(f) shows the various distributions for a given duration (600 s) or a given number of pyranometers (10). It confirms a stronger sensitivity to the number of measurement sites than to the integration time.

     In the previous series of tests, sensitivity to integration time was explored, although when the hourly distribution of SSI is the

target, there is no reason to integrate over a shorter period of time. However, when the instantaneous spatial distribution of SSI is sought, temporal integration can become a solution to construct the full distribution, at least when the cloud field is advected horizontally by the wind. This finding implies that we can trade a high spatial density of observations for longer integration times. Figure 5 documents how combining temporal and spatial sampling in the simulation allows the reconstruction of the reference instantaneous spatial distribution of SSI ($1280 \times 1280$ pyranometers at the center of the simulation hour). To retrieve

a distribution with an RMSD below $5 \times 10^{-4}$ compared to the reference, at least 10 pyranometers need to be deployed over 20 min, or alternatively 100 pyranometers over 5 min; the RMSD decrease is mainly controlled by that of the clear-sky mode (Fig. 5(b)). The retrieved distributions shown in Fig. 5(c) confirm that the reference is well captured with such measurement strategies. Interestingly, for long integration times, the RMSD starts increasing, which can be attributed to the non-stationarity of the SSI spatial field, although this non-stationarity does not result in RMSD exceeding $10^{-3}$ in this ideal simulation.

To summarize this sensitivity study combining observations and simulations, a minimum of 10 pyranometers, uniformly deployed over an area of roughly $10 \times 10 \text{ km}^2$ can capture the instantaneous SSI distribution when integrated for at least 10 min. This means that any such deployment meant to characterize cloud field properties can provide valuable information in a time resolution of roughly 10 minutes. This result aligns with the findings of Riihimaki et al. (2021), who observed that for hourly averages the bimodal distribution was challenging to identify from a single site but became much clearer when

cumulating data from 10 sites. The next step would be to propose a smart deployment strategy allowing to capture the SSI distribution with even fewer pyranometers or with a lesser RMSD. We did not fully address this question here, but report a few considerations. First, we observed in the few tests we did using the LES fields that it was possible to optimize the deployment

**Table 2.** Summary of the sensitivity tests. All units are in W m$^{-2}$, and correspond to normalized SSI, except for the last column (fraction of surface covered by shadow/clear sky) in percentage. Here, "shadow" refers to values less than 500 W m$^{-2}$, and "clear sky" to values greater than 900 W m$^{-2}$. p1 and p99 are respectively the first and $99^{th}$ percentiles of the distributions.

| Test case | RMSD total/shadow/clear sky ($\times 10^{-4}/\times 10^{-5}/\times 10^{-4}$) | mean total/shadow/clear sky | std total/shadow/clear sky | p1/p99 | fraction shadow/clear sky |
|---|---|---|---|---|---|
| Control | — / — / — | 961.2 / 293.9 / 1076.3 | 269.3 / 82.9 / 49.8 | 193.1 / 1214.3 | 12.2 / 82.6 |
| LWCx0.6 | 4.161 / 9.666 / 6.985 | 969.2 / 313.7 / 1065.2 | 240.7 / 84.5 / 49.6 | 210.6 / 1208.6 | 9.8 / 84.0 |
| LWCx1.4 | 2.181 / 7.006 / 3.628 | 954.9 / 281.5 / 1081.9 | 285.7 / 80.8 / 48.8 | 184.9 / 1213.7 | 13.7 / 81.7 |
| $\Delta$H = -400 | 7.228 / 10.640 / 12.208 | 961.0 / 321.0 / 1067.0 | 254.7 / 83.3 / 61.2 | 210.0 / 1261.1 | 11.5 / 83.0 |
| $\Delta$H = 400 | 3.933 / 7.358 / 6.627 | 961.6 / 280.3 / 1080.1 | 275.0 / 84.1 / 43.5 | 185.6 / 1187.0 | 12.4 / 82.4 |
| $\Delta$D = 400 | 6.700 / 14.749 / 11.186 | 961.1 / 336.5 / 1094.3 | 269.6 / 80.2 / 60.5 | 229.8 / 1242.7 | 12.0 / 77.1 |
| CC = 125 | 13.629 / 38.823 / 22.748 | 918.8 / 340.8 / 1136.8 | 341.2 / 70.5 / 74.3 | 236.4 / 1301.7 | 21.8 / 68.1 |

of $N$ pyranometers by selecting in an iterative way a combination of $N$ points that would minimize RMSD for a given field. This was done based on the knowledge of the full 2D SSI field hence can not, in practice, be repeated in a field experiment. In any case, the deployment that minimizes RMSD at a given time also generally yields larger or similar RMSDs than uniform deployment as close as 10 minutes away from that reference time. Hence, we believe that brute force optimization, even if it were possible in a true field experiment, would not be better than uniform. Nevertheless there might be a statistical distribution so that resulting RMSDs would be smaller than the uniform distribution for a large ensemble of cloud cases. This remains to be investigated. Although this sampling question has never been discussed for SSI to the best of our knowledge, it is a much more standard problem in the community of rain gauge deployment. The statistical tools developed by this community, in particular kriging, could be a source of inspiration for the future (Volkmann et al., 2010; Adhikary et al., 2015; Papamichail and Metaxa, 1996; Xu et al., 2018).

## 6 Sensitivity of SSI distributions to changes of cloud properties

To illustrate the sensitivity tests presented in Sect. 3.3, Figs. 6(a)-(b) show the vertical profiles of LWC and cloud fraction corresponding to one instant of the simulations ($51^{st}$ minute), for each sensitivity test. Figures 6(b)-(i) also show the simulated SSI fields at the same timestep, which further helps understand the modifications made to the cloud fields. Figure 7 and Table 2 present the results of the sensitivity tests in terms of the characteristics of the obtained distributions. In the following, the results are interpreted, with particular emphasis on the 3D effects; note that the highlighted mechanisms might differ for other SZA (here ranging from 34.5 to 36.8° only). We do not aim at providing an exhaustive analysis of Table 2; instead, we focus on a few mechanisms and discuss how we understand them, this understanding resulting from the combination of many available sources of information (prior theoretical and bibliographical knowledge, tables and figures presented in this work).

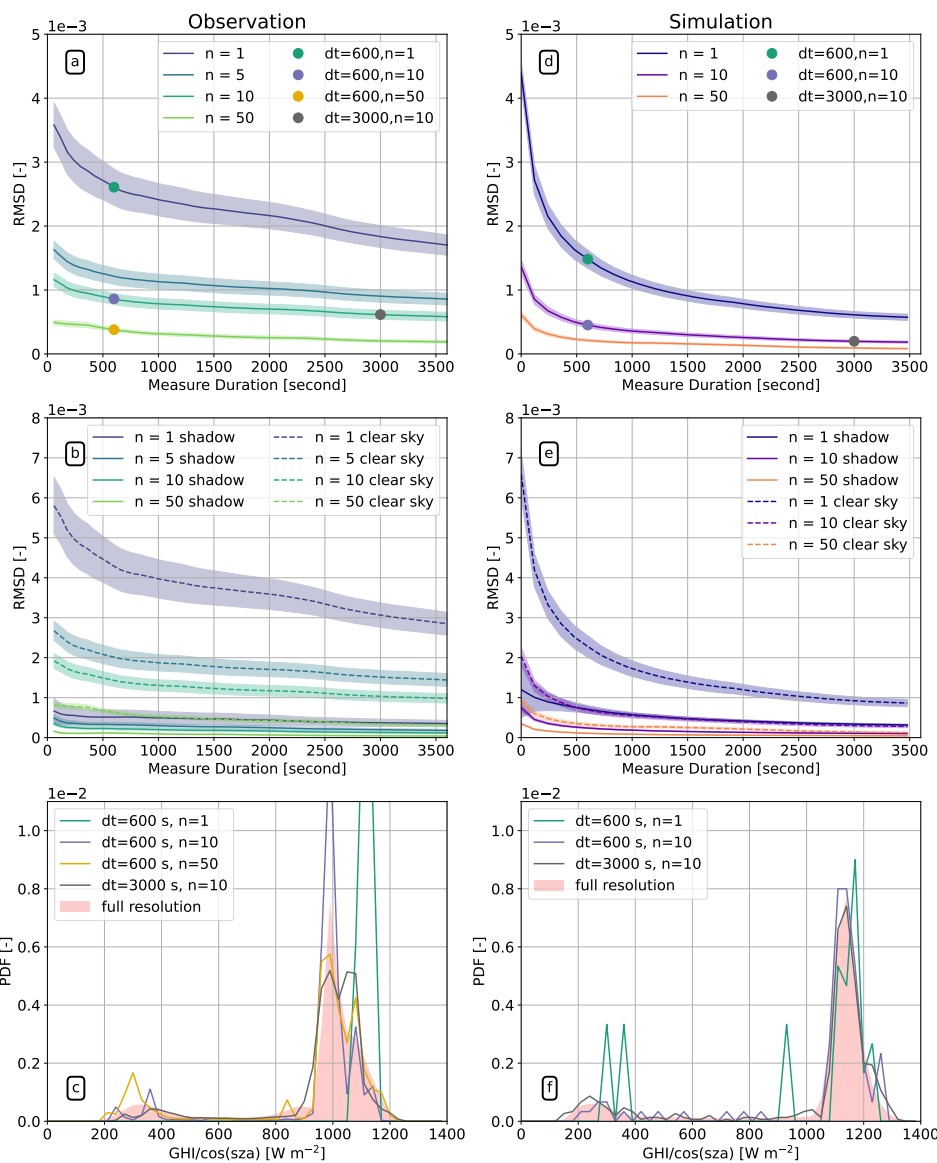

**Figure 4.** Spatiotemporal subsampling of the SSI distributions cumulated in observations (P5 period, left column) and simulation (Control, right column). Top row: RMSD with respect to the full distribution as a function of the number of points used for the computation and the cumulative period (x-axis) over which the distribution is computed; the shading indicates ± one standard deviation computed over 512 combinations for a given number of pyranometers and period. Middle row: RMSD computed over the cloud-shadow and the clear-sky modes. Bottom rows: examples of SSI distributions computed from (c) observations and (f) simulations for different numbers of pyranometers and periods; the shading represents the reference simulation computed over 95 pyranometers and cumulated over one hour for the observations and computed over 1280×1280 points and cumulated over one hour for the simulation.

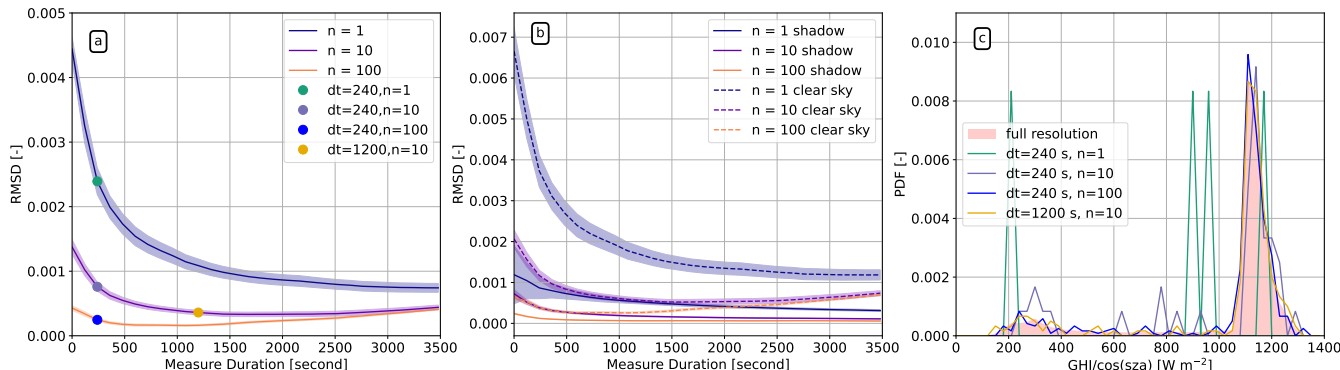

**Figure 5.** Strategy to measure instantaneous SSI distributions. (a) RMSD with respect to the mid-period instantaneous distribution for simulations as a function of the number of points used and the cumulative period over which the distribution is computed. The shading indicates ± one standard deviation computed over 512 combinations for a given number of points and period. (b) RMSD is computed over the cloud-shadow and the clear-sky modes. (c) Examples of SSI distributions retrieved from different numbers of measurement points. The shading represents the reference distribution computed over $1280 \times 1280$ points at a given instant.

First, we see that increasing cloud LWC reduces total mean SSI since at first order cloud reflectivity depends on the total water content in the field; more sunlight is reflected when clouds contain more water. Table 2 shows that increasing cloud LWC increases the shadow fraction (corresponding to pixels with SSI < 500 W m$^{-2}$ ) and reduces the clear-sky fraction, which is due to cloud edges being less transmissive than in the control simulation. Looking at the orange solid line in Fig. 7, we see that increasing LWC also shifts the cloud-shadow mode towards lower values and the peak in the clear-sky mode towards slightly larger values, which also results in a larger standard deviation of the total SSI (Table 2). This is due to a more widespread impact of reflection by cloud sides (as illustrated by the wider footprint of the white contours in Fig. 6(d) that materialize the 1100 W m$^{-2}$ isocontour), although the largest values (maximum illumination) remain the same as in the original field, suggesting a saturation of 3D effects with LWC. Reducing LWC (orange dashed line) results in the opposite effect.

Contrary to LWC, an increase in cloud base height alone does not change total mean SSI (Table 2). This is because, at first order, the mean SSI is driven by the cloud optical depth, which is unchanged. However, the SSI distribution (solid green line in Fig. 7) is sensitive to cloud base height: increasing cloud base height leads to an increase in the horizontal extension of the footprint of 3D effects (see Fig. 6(f)): as clouds are farther away from the surface, the downwelling diffuse flux can spread farther away from cloud sides before reaching the ground, which leads to more directly illuminated pixels being also affected by neighbouring clouds. Since the radiative flux is somehow diluted horizontally, the maximum illumination is smaller than for lower clouds. This also results in a smaller standard deviation of the clear-sky mode (Table 2). When clouds are closer to the surface (green dashed line in Fig. 7 and Fig. 6(g)), less clear-sky pixels are affected by clouds, but the very localized over-illumination by cloud sides is much more intense. Therefore, the mean radiative flux in the shadowed pixels is enhanced, and

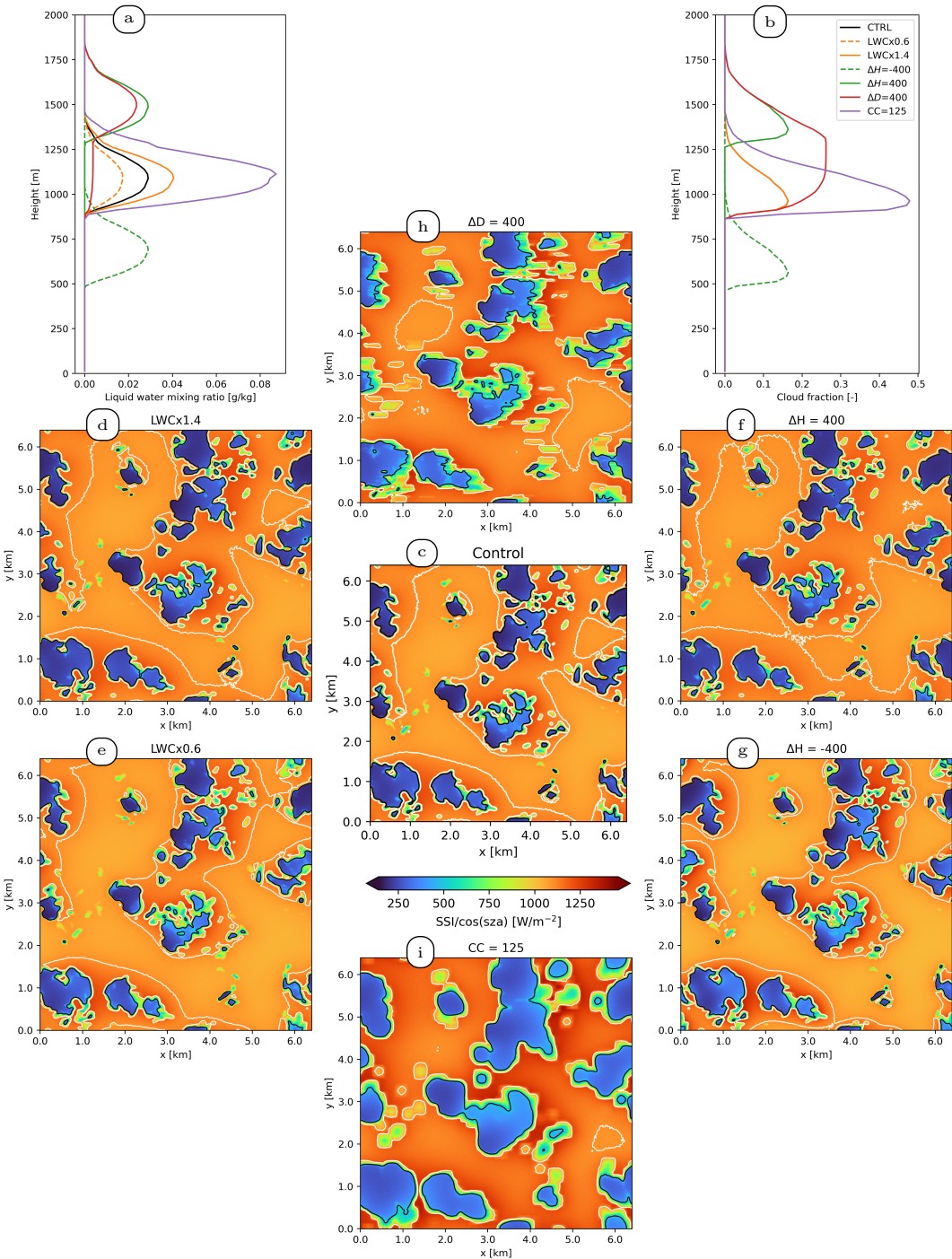

**Figure 6.** Cloud vertical profiles of (a) LWC and (b) cloud fraction, and SSI fields for control (c) and sensitivity tests (d-i) at minute 51 of the simulation. In the fields, colours represent SSI values, white lines represent the 1100 W m$^{-2}$ isocontour, and black lines represent the 500 W m$^{-2}$ isocontour.

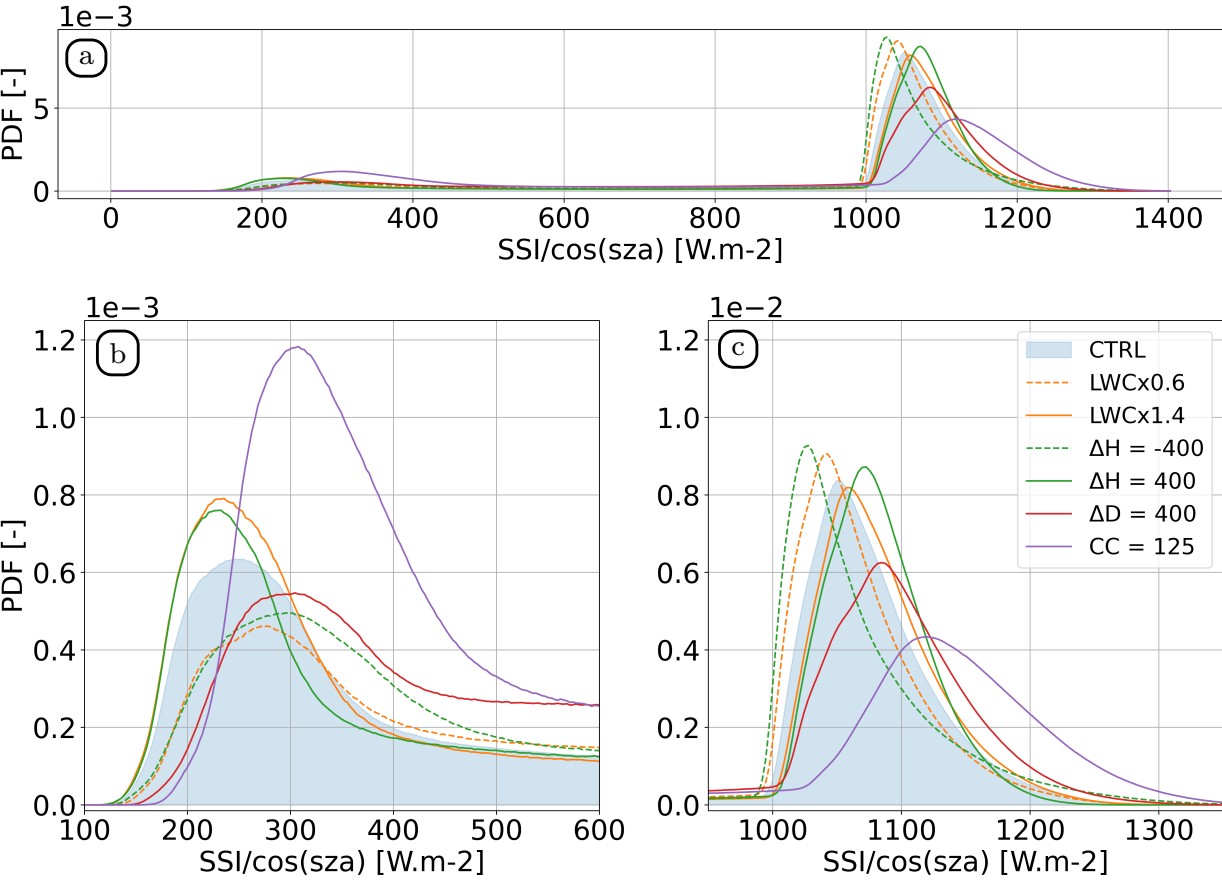

**Figure 7.** Simulated SSI distributions (a) and zooms over cloud-shadow (b) and clear-sky (c) modes for the sensitivity tests relative to cloud properties (LWC, cloud base height, cloud depth, and cloud cover). The blue-shaded distribution corresponds to the control simulation. Distributions were computed using all available data (1280 × 1280 pixels of size 5 m × 5m, × 60 minutes), using bins of 3 W m$^{-2}$. Details regarding the sensitivity tests are given in Table 1.

the shadow fraction is reduced because the shadow edges are illuminated. This again compensates and results in an unchanged mean SSI.

When the cloud layer depth is increased (red line in Fig. 7 and Fig. 6(h)), the fraction of shadowed pixels also increases since more radiation is intercepted by cloud sides (Várnai and Davies, 1999)). As more radiation is intercepted by cloud sides, 3D effects are more intense, and as this extra contribution comes from higher up in the atmosphere, they also have a wider footprint: for any clear-sky pixel, the fraction of visible sky that is occupied by bright, reflecting cloud sides increases with cloud geometrical depth. Shadows are also brighter (the mean radiative flux of the shadowed pixels is enhanced by more than 40 W m$^{-2}$), as clouds are overall optically thinner to slanted radiation because of the LWC scaling (necessary to preserve vertically integrated optical depth), and because more scattered radiation from the neighbouring clouds can reach the shadows

thanks to the wider scattering footprint. Note that this modification also leads to the conservation of mean SSI, but for different reasons than cloud height variations: for higher clouds, shadows are darker and clear-sky regions brighter; whereas for deeper clouds, both are brighter, but as clear-sky covers a smaller area of the surface, it compensates the overall right-shift of the pdf. Note that the shadow fraction is barely affected, meaning that mostly the proportion of intermediate values between 500 and 900 W m$^{-2}$ is increased. This holds true throughout the whole simulated hour (not shown); more extensive investigation should be performed to verify if this mean flux invariance is fundamental and remains true for other shifts of height and depth, as well as for other cloud types. Finally, an increase in cloud cover with conservation of the mean in-cloud LWC (hence increasing total LWC in the field) leads to wider shadows and more scattered radiation reaching the surface in clear-sky regions because cloud sides fill a larger portion of the sky which leads to more intense 3D effects, as demonstrated by the right-shift of the clear-sky mode (the distribution becomes dominated by large SSI values), similar to the case of deeper clouds. As wider clouds implies less space between neighbouring clouds, photons scattered by a given cloud side more easily reach a neighbouring cloud shadow, which in turn leads to brighter shadows compared to the control simulation.

Beyond the detailed modifications of SSI distributions discussed here, these sensitivity tests highlight that the impact of cloud geometrical and physical properties on SSI distributions results from various non-trivial 3D physical processes that make the interpretation much less straightforward than in the common plane-parallel framework. It is important to note that most of the tested modifications would have no significant effect on the SSI distribution under the independent column approximation of radiative transfer, where SSI distributions mostly depend on column-wise liquid water path and vertically projected cloud cover. The consideration of 3D effects is thus necessary to understand the modifications of the distributions and the way they relate to clouds. Beyond the mean SSI, its partition between the shadow and clear-sky modes is critical, as the nature of illumination (diffuse in cloud shadows vs mainly direct in clear-sky areas) makes a difference for many applications, in particular for PV production. Note also that due to strong non-linearities in the radiative transfer, the impact of combined modifications of the cloud field cannot be estimated by the linear combination of the impacts of the individual modifications, making this sensitivity study primarily useful for the qualitative understanding of the impacts rather than their quantitative assessment in real situations.

## 7   Conclusions

In this paper, we focused on the instantaneous SSI spatial distribution, an under-explored quantity that is of utmost importance for surface-atmosphere interactions and solar energy applications, especially under cumulus cloud conditions. We investigated the spatial distribution from both an observational and a numerical perspective. Spatially dense SSI observations from the HOPE field campaign constitute a unique resource to investigate this otherwise barely accessible quantity. By comparing the observed distributions for carefully selected cumulus situations to those simulated with state-of-the-art cloud and 3D radiative transfer modelling, we showed that the numerical simulations are sufficiently reliable to further explore the links between cloud field properties and SSI distributions.

We then investigated how the instantaneous SSI spatial distributions can be estimated using a limited number of pyranometers by taking advantage of cloud motion that allows the sampling of a stationary cloud field from fixed points at the surface. We demonstrated that, for a $10 \times 10$ km$^2$ area, cumulating observations from 10 pyranometers over 10 min can provide the same information on SSI spatial distribution as using 100 pyranometers over a shorter time period. Preliminary tests of optimizing the spatial distribution of the pyranometers also indicated that 15 optimally distributed pyranometers could capture the same spatial variability as 100 uniformly distributed pyranometers. However, this deserves further analysis to understand how such an optimized network can be deployed when one does not know in advance the details of the fields that will be observed. Although the measurement strategy investigation was limited to cumulus situations and did not consider the impact of aerosols, which could affect the results, this preliminary study was meant to demonstrate how simulations can be used to address this question. It certainly provides a valuable basis for further dedicated, more detailed studies, and paves the way for designing measurement strategies tailored for specific applications related to the high-resolution characterization of SSI.

The simulation system was also used to study the sensitivity of the SSI distribution to the cloud properties. This highlighted that both the geometrical and physical properties of the clouds can alter the SSI distribution, via the combination of complex physical processes which are sometimes hard to disentangle. We nevertheless tried to emphasize that the irradiance at some locations results from the contributions of the blue sky, the cloud edges and the cloud bottoms, which are combined according to their respective proportions in the hemisphere and luminance. This sensitivity study is again somehow preliminary and would deserve a dedicated study, allowing to explore a variety of cumulus fields, not to mention other cloud types. Importantly such future work should check whether the sensitivities highlighted by the simulations can be identified in the observations. For that purpose the HOPE dataset, gathering several remote sensing instruments, would be very relevant.

As pointed out before, we did not consider aerosols and instead focused on the impact of clouds only. In reality, aerosols are ubiquitous, but their detailed representation in atmospheric and radiative models is challenging because their optical properties depend on their size and composition, but also on their hygroscopicity and ambient humidity. Several physical processes have also been identified that can explain the increase of aerosol optical depth in the vicinity of cumulus clouds (Eck et al., 2014). Besides this complexity, we did not have observational data to properly account for their effects. Although aerosols were not accounted for in the simulations we believe that the qualitative results of the paper, along with the physical interpretations regarding the impact of cloud properties, would largely hold for real conditions. Yet, it is useful to discuss what impacts aerosols would have. According to Gristey et al. (2022), the presence of aerosols would typically shift the clear-sky mode to lower values due to increased absorption and the cloud-shadow mode to higher values due to extra radiation scattered laterally towards cloud shadows. Quantitatively, this impact could be as significant as those obtained from the sensitivity tests. In any case, accounting properly for aerosols would have required appropriate observations and a detailed optical module that were beyond the scope of the present paper. In view of making our understanding of the characteristics of SSI distributions and their sensitivity to the overlaying atmosphere more exhaustive, future work should strive to include aerosols in the simulations and perform additional sensitivity tests.

This work is meant to be exploratory and to highlight a poorly known quantity that we believe will become of much more interest to the research community in the coming years as the resolution of numerical weather prediction models increases

and as observational capabilities for characterizing 3D cloud structures improve. This study confirms that the SSI distribution contains valuable information on cloud properties, including its 3D geometrical properties that most cloud profiling instruments cannot fully capture due to limited spatial sampling. Future work should thus focus on the derivation of relevant cloud properties from a network of pyranometers, which would be a significant step forward for atmospheric sciences. For cumulus situations, the mean and standard deviation of the two peaks of the bimodal SSI distribution seem to provide a wealth of yet unexplored information. This study could also be extended to other campaigns conducted with the pyranometer network. This includes the HOPE dataset acquired near Melpitz, where the network was deployed in a much smaller area of roughly $500 \times 500$ m$^2$, to investigate variability at even smaller scales. Recently, the network was deployed in the framework of the Small-Scale Variability of Solar Radiation campaign (S2VSR), which was conducted at the ARM Southern Great Plains site and targeted an area of 6 x 6 km$^2$. Although the dataset was not yet available at the start of this investigation, several ancillary observations are available based on routine ARM measurements, which can help further understand the factors influencing the SSI distribution. Calibration/validation campaigns are also planned for the upcoming launch of the EarthCARE satellite mission, where small-scale radiative closure experiments will be carried out and would benefit from such an instrumental deployment.

The fact that SSI distributions are so tightly related to most 3D thermodynamical properties of the atmosphere also offers an advanced framework for evaluating LES in a much more stringent way than currently done when LES properties are generally spatially averaged to be compared to vertical profiles at well-instrumented sites. In particular, we believe that the correct representation of LWC heterogeneity, which currently represents a challenge for LES, could be tackled with such observations. Relying on the objective determination of cumulus cloud conditions set up in this study, we also advocate the development of cloud classifications based on SSI observations using the metrics introduced in this study, in line with the random-forest classifier recently proposed by Sedlar et al. (2021). Such classifications could be used for comparison of cloud conditions at different sites or to study the variability of weather conditions in a much more robust way than human-based classifications. These diverse perspectives highlight the potential of considering SSI spatial distributions and suggest that in the future, networks of radiation sensors should be more systematically deployed during field campaigns dedicated to boundary layer clouds and surface-atmosphere interactions.

**Data availability**

Simulation data supporting our results are available on Zenodo (He et al., 2024). The repository includes radiative transfer simulation outputs, scripts to launch radiative transfer simulations with htrdr version 0.8.1 (https://www.meso-star.com/projects/htrdr/htrdr.html, source code also in the archive) and to reproduce tables and figures, as well as namelists to run Large-Eddy Simulations with the community code Meso-NH, version 5.4.3 (http://mesonh.aero.obs-mip.fr/mesonh/dir_open/dir_MESONH/MNH-V5-4-3.tar.gz). Observational data are available at re3data (Registry of Research Data Repositories., 2017; Bomidi, 2022).

**Author contribution**

ZH, NV, QL and FC analysed the data; NV, FC and ZH performed the simulations; QL, FC, NV and ZH wrote the manuscript draft; HD and JW reviewed and edited the manuscript.

**Competing interests**

The authors declare that they have no conflict of interest.

*Acknowledgements.* The research leading to this work has been carried out as a part of the Smart4RES project (European Union's Horizon 2020, No. 864 337). Part of this work was carried out in the framework of the Fine4Cast project, funded by France 2030 (ANR reference: 22-PETA-0008).

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
