# Peer review of "How to observe the small-scale spatial distribution of surface solar irradiance, and how is it influenced by cumulus clouds?"

_EGUsphere, 2024_

## Referee Comment (RC1)

**Review of "How to observe the small-scale spatial distribution of surface solar irradiance, and how is it influenced by cumulus clouds?" by He et al.**

**15 May 2024**

**General comments**

This study provides an analysis of observed and simulated surface solar irradiance (SSI) variability over land at small spatial scales for broken cloud conditions. After establishing the realism of the simulations, several sensitivity tests are applied to the simulations to explore the impact of cloud properties on the SSI variability. It is also proposed that a dense network of 10 surface-based pyranometers integrated over 10 mins can capture the main details of the SSI variability, providing useful guidance for sampling strategies targeted at future measurement campaigns.

The manuscript is very well written. Figures are mostly clear. The references are mostly appropriate. I feel that the features of SSI variability that are discussed are largely already known and documented in other the recent literature that has combined LES of shallow cumulus clouds with 3D radiative transfer. However, the observational focus of this study provides a somewhat different perspective that I think will still be of interest to the community. There are a few areas that would benefit from some further clarification and/or discussion, as outlined in my comments below. After addressing these minor comments, I recommend that the study be considered for publication in ACP.

**Specific comments**

L20-22: It is a common misconception that 3D cloud effects vanish with spatial and temporal averaging. Systematic 3D biases can remain after averaging. For example, see Fig. 6a in Gristey et al., 2020b (already cited in the submitted paper).

Section 2.1: It would be helpful to include some basic statistics such as the min/max/mean distances between the pyranometers. This will help with comparisons to the LES grid.

L133: The spatial resolution of the 3DRT seems to be much finer than the LES, is that correct? If so, relatively coarsely resolved clouds are contributing to relatively finely resolved surface irradiance. Can a justification be provided that this is physically reasonable?

Section 2.3: Are aerosols ignored in the simulations? It has been shown that aerosols, which are always present in reality to some extent, can have a substantial impact on the SSI PDF in such conditions (https://doi.org/10.1029/2022JD036822). This should at least be noted in the manuscript as a potential source of error assuming that aerosols are indeed excluded. It is interesting that the clear-sky PDFs (Fig. 3b) match quite well between the observations

and simulations even without aerosols in the simulations. Perhaps this was a very pristine day. I recommend adding some discussion on this topic.

L177-178: The all sky images are a good check but are somewhat limited in spatial extent. The authors could also check satellite images from those days to confirm the presence of a widespread shallow cumulus cloud field at the location of interest.

L214: The choice of 500 W/m2 and 900 W/m2 seems rather arbitrary. Also, since the location of the two modes can vary from case to case or with different cloud properties (for example see Fig. 7), using fixed thresholds seems like it would incur some error because different portions of the PDFs will be represented. The authors could consider adjusting these thresholds depending on the PDF shape for a given case. Otherwise, I think a short discussion of this limitation in the manuscript is required.

L221-222: This approach assumes that the impact of different cloud properties on SSI can be linearly combined. Non-linearities are inherent in radiative transfer, especially 3D radiative transfer, so this might not be a great assumption. It should be noted that non-linearities are not accounted for, unlike the machine learning approaches used in other studies that are already mentioned.

L254-255: I expect the enhancement increase with cloud cover is because the remaining clear-sky region is receiving scattered radiation from more surrounding clouds.

Figure 3: The figure caption does not match what is plotted. I think b&c are swapped with d&e in the caption. I recommend revising the caption to improve clarity.

Section 5: The discussion and conclusions from this section are broadly consistent with that found in Fig. 7 here: https://doi.org/10.1175/BAMS-D-19-0227.1. They found that, when averaging for 1 hour, the bimodal PDF was difficult to identify from a single site, but became much clearer when considering 10 sites. I recommend including this relevant comparison.

L327: Cloud lifetime is only about 15mins for these clouds so the Taylor hypothesis probably does not hold very well in this environment.

L371-372: As well as the optically-thinner slant path, the wider footprint of scattering also means one cloud can be scattering more radiation into the shadow of a neighboring cloud, therefore adding to the "brightening".

L422: The suggestion that SSI observations could provide a constraint for LES cloud droplet effective radius heterogeneity is not related to any of the sensitivity tests performed in this study. It's an interesting suggestion but it comes a bit out-of-the-blue. If the authors would like to keep this statement, I think another sentence or two is required to provide some physical reasoning to support this claim.

L424-426: Cloud classifications that utilize SSI observations are already being developed. An appropriate reference to cite here is here: https://doi.org/10.1175/JAMC-D-20-0153.1.

L434-435: *"It does not include Large Eddy Simulation 3D output fields but they will be provided on demand."* I am not sure that a statement like this fits well with the journal data policy. Since the authors are providing the data needed to produce the plots, and the underlying tools are available (correct?), it might be best to remove this statement.

---

## Author Response (AR1)

**Response to the reviewers**

We thank the reviewers for the critical assessment of our work. Below is our response point by point. The comments by the reviewer are indicated in *italic*. The changes made to the manuscript are detailed in red. The referenced lines correspond to the updated version.
* * *
**Reviewer 1**

*This study provides an analysis of observed and simulated surface solar irradiance (SSI) variability over land at small spatial scales for broken cloud conditions. After establishing the realism of the simulations, several sensitivity tests are applied to the simulations to explore the impact of cloud properties on the SSI variability. It is also proposed that a dense network of 10 surface-based pyranometers integrated over 10 mins can capture the main details of the SSI variability, providing useful guidance for sampling strategies targeted at future measurement campaigns. The manuscript is very well written. Figures are mostly clear. The references are mostly appropriate. I feel that the features of SSI variability that are discussed are largely already known and documented in other recent literature that has combined LES of shallow cumulus clouds with 3D radiative transfer. However, the observational focus of this study provides a somewhat different perspective that I think will still be of interest to the community. There are a few areas that would benefit from some further clarification and/or discussion, as outlined in my comments below. After addressing these minor comments, I recommend that the study be considered for publication in ACP.*

**Reply**: We thank the reviewer for his/her positive assessment of our work. As pointed out in the introduction and stressed by the reviewer, the main contribution of this work with regards to the recent literature on the subject is the extensive use of observations from a dense network of pyranometers. However we also believe that the sensitivity studies (and related interpretations) based on the modified LES fields, as well as the preliminary discussion on instrument deployment, are complementary to what has been done before. In the following, we address the reviewer's concerns point by point.

**Reviewer Point P 1.1** — *L20-22: It is a common misconception that 3D cloud effects vanish with spatial and temporal averaging. Systematic 3D biases can remain after averaging. For example, see Fig. 6a in Gristey et al., 2020b (already cited in the submitted paper).*

**Reply**: We thank the reviewer for this clarification. The corresponding sentence was adjusted as follows at l.20: "[...] this cloud enhancement has not been much investigated in the atmospheric science community, primarily because it is thought to vanish with spatial and temporal averaging on scales relevant to energetic transfers in the Earth system, even though recent work has demonstrated that systematic biases could remain even on daily averages (Gristey et al., 2020)".

**Reviewer Point P 1.2** — *Section 2.1: It would be helpful to include some basic statistics such as the min/max/mean distances between the pyranometers. This will help with comparisons to the LES grid.*

**Reply**: We thank the reviewer for this suggestion. The information is added as follows at l.106:
    "The minimum and maximum distances between any two pyranometers are 0.14 and 14.1 km, and the mean distance to the closest neighbour is 0.86 km"

**Reviewer Point P 1.3** — *Are aerosols ignored in the simulations? It has been shown that aerosols, which are always present in reality to some extent, can have a substantial impact on the SSI PDF in such conditions (https://doi.org/10.1029/2022JD036822). This should at least be noted in the manuscript as a potential source of error assuming that aerosols are indeed excluded. It is interesting that the clear-sky PDFs (Fig. 3b) match quite well between the observations and simulations even without aerosols in the simulations. Perhaps this was a very pristine day. I recommend adding some discussion on this topic.*

**Reply**: This is an important point also raised by Reviewer 2. Indeed, the simulations have been performed without any aerosols, and this is now clearly stated l.148:

"Importantly, the simulations are performed without aerosols, although they can significantly alter the SSI distribution (Gristey et al., 2022)."

The potential impact of aerosols is now explicitly mentioned in the discussion (l.465):

"As pointed out before, we did not consider aerosols and instead focused on the impact of clouds only. In reality, aerosols are ubiquitous, but their detailed representation in atmospheric and radiative models is challenging because their optical properties depend on their size and composition, but also on their hygroscopicity and ambient humidity. Several physical processes have also been identified that can explain the increase of aerosol optical depth in the vicinity of cumulus clouds (Eck et al., 2014). Besides this complexity, we did not have observational data to properly account for their effects. Although aerosols were not accounted for in the simulations we believe that the qualitative results of the paper, along with the physical interpretations regarding the impact of cloud properties, would largely hold for real conditions. Yet, it is useful to discuss what impacts aerosols would have. According to Gristey et al. (2022), the presence of aerosols would typically shift the clear-sky mode to lower values due to increased absorption and the cloud-shadow mode to higher values due to extra radiation scattered laterally towards cloud shadows. Quantitatively, this impact could be as significant as those obtained from the sensitivity tests. In any case, accounting properly for aerosols would have required appropriate observations and a detailed optical module that are beyond the scope of the present paper. In view of making our understanding of the characteristics of surface flux distributions and their sensitivity to the overlaying atmosphere more exhaustive, future work should strive to include aerosols in the simulations and perform additional sensitivity tests".

Besides these clarifications on the impact of aerosols, we have performed additional 3DRT simulations to study the impact of aerosols on the SSI pdf. The resulting simulations are the following:

- Control: without aerosol

- Case 0a: homogeneous concentration from the surface to cloud top such that total optical thickness $\approx 0.1$ (which is close to the similar study in Gristey et al. (2022))

- Case 0b: homogeneous concentration from the surface to cloud top such that optical thickness $\approx 0.15$

- Case 1: homogeneous concentration from the surface to cloud base, then 0 above, such that optical depth $\approx 0.1$

- Case 2: homogeneous concentration from the surface to cloud base such that optical thickness $\approx 0.1$ and homogeneous in the cloud layer such that optical thickness $\approx 0.05$. Total optical thickness $\approx 0.15$

- Case 3: heterogeneous concentration which is proportional to the concentration of water vapour such that optical thickness $\approx 0.1$ below the cloud base and optical thickness $\approx 0.05$ in the cloud layer

Figure 1 shows the results of these simulations, in terms of SSI distributions and fields. It confirms that the aerosol optical depth plays an important role in shifting the two peaks of the SSI distribution. The clear peak is shifted to the left and the shadow peak is shifted to the right, correlated to what is found in Gristey et al. (2022). The aerosols below the clouds (Case 1) have more impact than those within the clouds (Case 0a). In contrast, the spatial variability of aerosol seems to have less impact on SSI pdf. These results suggest that the overall shape of the SSI distribution is not altered by the presence of aerosols, so our interpretations of the Control case remain relevant for real conditions.

Regarding the satisfactory match between the observed and clear-sky SSI pdfs, it probably comes from the fact that the conditions were relatively clean :

"They are both unimodal and symmetric, with approximately the same width and around the same mean value, suggesting that the impact of aerosols, which are not accounted for in the simulations, was rather limited for that particular day." (at l.271)

Note also that in dry conditions, the impact of aerosols is modest, compared to the very large AOD enhancement reported in the vicinity of clouds, and also simulated in Gristey et al. (2022) due to their hygroscopic growth. Hence the impact of aerosols is expected to be much larger in the presence of broken clouds due to what happens near cloud edges.

**Reviewer Point P 1.4** — *L133: The spatial resolution of the 3DRT seems to be much finer than the LES, is that correct? If so, relatively coarsely resolved clouds are contributing to relatively finely resolved surface irradiance. Can a justification be provided that this is physically reasonable?*

**Reply**: The spatial resolutions of the 3DRT and LES are indeed different, the 3DRT resolution being much finer (5 m vs 25 m). Actually, even coarse clouds can result in fine features at the surface when considering rays passing near cloud edges, for instance, those contributing to cloud enhancement. Hence, to fully resolve the intensity and precise location of cloud enhancement, increasing the 3DRT resolution is crucial. To support this statement, Fig. 2 shows two simulated SSI maps and Fig. 3 shows the corresponding distribution with two 3DRT resolutions: 25 m (same as LES) and 5 m. Small differences can be noticed near cloud shadows' edges, and the distribution becomes smoother at a finer resolution. A short explanation is added at l.139:

"Each pixel of each field is a $5 \times 5$ m$^2$ square. Note that a finer resolution than the LES is used to accurately simulate what happens near cloud shadow edges, where variations occur at smaller scales than the cloud resolution. Such a fine resolution allows to correctly simulate the rapid transition from the shadow to the clear-sky areas, and to capture the value of the maximum cloud enhancement, which is essential to reproduce the SSI distribution."

**Reviewer Point P 1.5** — *L177-178: The all sky images are a good check but are somewhat limited in spatial extent. The authors could also check satellite images from those days to confirm the presence of a widespread shallow cumulus cloud field at the location of interest.*

**Reply**: We thank the author for this kind advice. We did not find satellite images that precisely match the "golden cases" selected in our study. However, we did find a few satellite images with some spatial and temporal deviations (see Fig. 4). The information about the exact time when these satellite images are taken is lost in the database (Sentinel-Hub), but it can be inferred from Heinze et al. (2017).

[Figure]

Figure 1: a preliminary study of aerosol's impact on the SSI pdf. (a) Vertical aerosol profile. (b) The SSI pdf with the presentation of aerosols. (c - i) The flux maps.

[Figure]

Figure 2: SSI map with different 3DRT resolution. Left: 25m. 3DRT uses the same resolution than it is used for LES. Middle: 5m. Right: Difference of the two SSI maps.

[Figure]

Figure 3: SSI pdf with different 3DRT resolution. blue: 25m. 3DRT uses the same resolution than it is used for LES. orange: 5m.

All satellite images are taken a few minutes to a few hours before the selected periods in this study (see caption of Fig. 4 for more details). Cumulus clouds can be clearly seen in P1, P3, P4 and P5 (Fig. 4(a,c-f)), but is not clearly shown in Fig. 4(b) for P2. Referring to Fig. 3 in the manuscript, P2 has a lower cloud fraction and lower SSI enhancement than other periods. We assume that the cumulus clouds at this period are smaller than the resolution of MODIS Terra ($1 \times 1$ km$^2$), which is why they are not captured in the satellite image. This confirmation of cumulus presence is now specified ( l.189): "Thanks to the all-sky images and to MODIS satellite images, it was verified that they indeed correspond to cumulus cloud situations, thereby validating our automatic selection procedure.

**Reviewer Point P 1.6** — *L214: The choice of 500 W/m2 and 900 W/m2 seems rather arbitrary. Also, since the location of the two modes can vary from case to case or with different cloud properties (for example see Fig. 7), using fixed thresholds seems like it would incur some error because different portions of the PDFs will be represented. The authors could consider adjusting these thresholds depending on the PDF shape for a given case. Otherwise, I think a short discussion of this limitation in the manuscript is required.*

**Reply**: We thank the reviewer for this comment. This limitation is now pointed out at l.228:
   "Although these two thresholds are arbitrary, the main objective was to qualitatively isolate both modes, which proved to be acceptable for the cases encountered. However, defining both modes in a more flexible way, which would depend on the actual distribution and would work for a larger variety of cloud properties, would be useful and should be considered for future studies."

**Reviewer Point P 1.7** — *L221-222: This approach assumes that the impact of different cloud properties on SSI can be linearly combined. Non-linearities are inherent in radiative transfer, especially 3D radiative transfer, so this might not be a great assumption. It should be noted that non-linearities are not accounted for, unlike the machine learning approaches used in other studies that are already mentioned.*

[Figure]

(a) MODIS Aqua, 2013-04-18

(b) MODIS Terra, 2013-04-20

(c) MODIS Aqua, 2013-04-25

(d) Landsat8-9, 2013-04-25, 10:23

(e) MODIS Terra, 2013-05-05

(f) MODIS Aqua, 2013-05-05

Figure 4: The satellite images that are closed to the selected periods in our study in UTC from Sentinel-Hub: P1: 18 April 2013, 13:00-14:00; P2: 20 April 2013, 9:12-10:12; P3: 25 April 2013, 12:32-13:32; P4: 5 May 2013, 9:30-10:30; P5: 5 May 2013, 11:36-12:36. The blue contour indicates the pyranometer network. It is noted that the exact time when the images of MODIS Terra and Aqua were taken was missing in the database. However, we presume those images of MODIS Terra were taken around 11:30-12:45 and images of MODIS Aqua were taken around 8:30-9:30 based on Heinze et al. (2017).

**Reply**: We thank the reviewer for this clarification. It is true that the total response will not be the linear combination of individual responses, due to obvious non-linearities. This sensitivity study primarily aims to provide physical insight into how each cloud parameter influences the SSI. This is now more clearly stated l.234:

"Sensitivity tests are performed in Sect. 6 in order to gain physical insight on how various cloud characteristics drive SSI distributions. For each category of test, the 60 LES cloud fields of the one-hour-long simulation are modified, varying a single property at a time, among cloud LWC, cloud base height, cloud depth, or cloud fraction."

and l.434:

"Note also that due to strong non-linearities in the radiative transfer the impact of combined modifications of the cloud field cannot be estimated by the linear combination of the impacts of the individual modifications, making this sensitivity study primarily useful for the qualitative understanding of the impacts rather than their quantitative assessment in real situations."

**Reviewer Point P 1.8** — *L254-255: I expect the enhancement increase with cloud cover is because the remaining clear-sky region is receiving scattered radiation from more surrounding clouds.*

**Reply**: We thank the reviewer for this suggestion. It is added at l.269:

"We believe that it is because the clear-sky region is receiving scattered radiation from more surrounding clouds (the sensitivity of the SSI distribution to cloud cover is investigated in Sect. 6)."

**Reviewer Point P 1.9** — *Figure 3: The figure caption does not match what is plotted. I think b and c are swapped with d and e in the caption. I recommend revising the caption to improve clarity.*

**Reply**: We thank the reviewer for this careful comment. It is corrected in the revised version.

**Reviewer Point P 1.10** — *Section 5: The discussion and conclusions from this section are broadly consistent with that found in Fig. 7 here: https://doi.org/10.1175/BAMS-D-19-0227.1. They found that, when averaging for 1 hour, the bimodal PDF was difficult to identify from a single site, but became much clearer when considering 10 sites. I recommend including this relevant comparison.*

**Reply**: We thank the reviewer for pointing to this reference, which is now mentioned at l.363:

"This result aligns with the findings of Riihimaki et al. (2021), who observed that for hourly averages the bimodal distribution was challenging to identify from a single site but became much clearer when cumulating data from 10 sites."

**Reviewer Point P 1.11** — *L327: Cloud lifetime is only about 15mins for these clouds so the Taylor hypothesis probably does not hold very well in this environment.*

**Reply**: We agree with the reviewer and the Taylor hypothesis is removed in the revised version.

**Reviewer Point P 1.12** — *L371-372: As well as the optically-thinner slant path, the wider footprint of scattering also means one cloud can be scattering more radiation into the shadow of a neighbouring cloud, therefore adding to the "brightening".*

**Reply**: We agree with the reviewer on this point and it is added at l.411:

"Shadows are also brighter, as clouds are overall optically thinner to slanted radiation because of the LWC scaling, and because more scattered radiation from the neighbouring clouds can reach the shadows thanks to the wider scattering footprint."

**Reviewer Point P 1.13** — *L422: The suggestion that SSI observations could provide a constraint for LES cloud droplet effective radius heterogeneity is not related to any of the sensitivity tests performed in this study. It's an interesting suggestion but it comes a bit out-of-the-blue. If the authors would like to keep this statement, I think another sentence or two is required to provide some physical reasoning to support this claim.*

**Reply**: Indeed, the sensitivity studies performed in this paper are with respect to the liquid water content and cloud geometry. The heterogeneity of the droplet effective radius in LES is not explored these sensitivity studies. The corresponding statement is removed in the revised version.

**Reviewer Point P 1.14** — *Cloud classifications that utilize SSI observations are already being developed. An appropriate reference to cite here is here: https://doi.org/10.1175/JAMC-D-20-0153.1.*

**Reply**: We thank the reviewer for this bibliographic complement. It is added at l.498:
"Relying on the objective determination of cumulus cloud conditions set up in this study, we also advocate the development of cloud classifications based on SSI observations using the metrics introduced in this study, in line with the random-forest classifier recently proposed by Sedlar et al. (2021)."

**Reviewer Point P 1.15** — *L434-435: "It does not include Large Eddy Simulation 3D output fields but they will be provided on demand." I am not sure that a statement like this fits well with the journal data policy. Since the authors are providing the data needed to produce the plots, and the underlying tools are available (correct?), it might be est to remove this statement.*

**Reply**: We thank the reviewer for this careful comment. As mentioned, the LES simulation results can be obtained by running Meso-NH with the name lists provided in He et al. (2024) on Zenodo. The statement on LES (un)availability has been removed accordingly.
* * *
**Reviewer 2**

**Reviewer Point P 2.1** — *The paper contains many interesting insights in an interesting topic. The paper however could be much improved in its presentation, as it is at several points unclear in its goals (I will explain below), and also contains many interesting results in the figures and tables that are not explained in the text. Also, the observational strategy is not generic, as it only applied to cumulus, it would be nice of some deeper reflection on other cloud types is added to the paper.*

**Reply**: We thank the reviewer for the comments and suggestions. First, the original title of the paper, *How to observe the small-scale spatial distribution of surface solar irradiance, and how is it influenced by cumulus clouds*, was indeed misleading, because the focus on cumulus clouds was somehow unclear, and more importantly because it was too much focused on the measurement strategy, which is only one component of the presented work. This study does not aim to provide a universal observational

strategy for SSI variability, of which the sources are diverse (cloud organization, heterogeneous aerosols and surfaces, etc.), it rather illustrates how simulations can be used to meet this target. In fact, this study more broadly investigates the impacts of cumulus clouds on the SSI variability, combining unique observations and numerical sensitivity studies. We focus on cumulus clouds since they are critical for local SSI variability and have already been the single focus of many previous studies on this topic. This variability occurs at fine spatial and temporal scales, which has influences on small-scale convection, urban thermal studies and especially PV production. These small scales are also poorly resolved by numerical weather prediction models and other models relying on 1D physical parameterizations, making them a challenging target.

To gain in clarity, the following changes are made:

- The title is changed into something more generic, which better encompasses all the questions addressed therein: Combining observations and simulations to investigate the small-scale variability of surface solar irradiance under continental cumulus clouds.

- We do not pretend to develop a universal observation strategy because the cumulus field to be observed can be very variable, and the aerosols (not accounted for in the study) certainly play an important role (as pointed out by both reviewers and discussed carefully below). Deriving a more general measurement strategy would probably deserve a dedicated study and is let for a future paper. Nonetheless, we believe that the qualitative result that SSI variability caused by cumulus clouds can be captured using 10 pyranometers integrated for 10 minutes is a preliminary useful information. This is clarified in the conclusion at l.453:

"Although the measurement strategy investigation was limited to cumulus situations and did not consider the impact of aerosols, which could affect the results, this preliminary study was meant to demonstrate how simulations can be used to address this question. It certainly provides a valuable basis for further dedicated, more detailed studies, and paves the way for designing measurement strategies tailored for specific applications related to the high-resolution characterization of SSI."

In fact, we attempted to optimize the pyranometer deployment for a period of time using the simulated cumulus cloud field, but we found that the optimized deployment for a period is not universal (other periods have different optimized deployments). We found some inspiring studies on optimizing rain gauge deployment using the kriging technique, but we think that universal sampling strategies for SSI variation are a tricky subject that needs further research. This is clarified at l.365:

"The next step would be to propose a smart deployment strategy allowing to capture the SSI distribution with even fewer pyranometers or with a lesser RMSD. We did not to address this question here, but report a few considerations. First, we observed in the few tests we did using the LES fields that it was possible to optimize the deployment of $N$ pyranometers by selecting in an iterative way a combination of $N$ points that would minimize RMSE for a given field. This was done based on the knowledge of the full 2D SSI field hence can not, in practice, be repeated in a field experiment. In any case, the deployment that minimizes RMSD at a given time also generally yields larger or similar RMSDs than uniform deployment as close as 10 minutes away from that reference time. Hence, we believe that brute force optimization, even if it were possible in a true field experiment, would not be better than uniform. Nevertheless there might be a statistical distribution so that resulting RMSDs would be smaller than the uniform distribution for a large ensemble of cloud cases. This remains to be investigated. Although this sampling question has

never been discussed for SSI to the best of our knowledge, it is a much more standard problem in the community of rain gauge deployment. The statistical tools developed by this community, in particular kriging, could be a source of inspiration for the future (Volkmann et al., 2010; Adhikary et al., 2015; Papamichail and Metaxa, 1996; Xu et al., 2018) "

- The focus on cumulus clouds is now better motivated (l.81), but we do not extend to other cloud types because the corresponding scientific issues (e.g. horizontal heterogeneity, multiple layers, etc.) can be very different than those raised in this paper and cannot be efficiently mentioned in a few sentences only:

  "In line with previous studies addressing this question we focus on cumulus clouds because they are responsible for the largest small-scale variability of SSI. These clouds, ubiquitous across a large fraction of the globe, also remain a challenge for weather and climate modelling, primarily because their small size means that they are generally parameterized, and their radiative impact as well."

**Reviewer Point P 2.2** — *The paper is in fact a two part paper, first it explains ideas and theory on how to observe the detailed spatial distribution of surface solar irradiance, and in the second part details about radiation under cumulus fields are explained. It got a little bit confused while reading the second part, as I expected there that an elaborate study would be presented on how to optimally observe radiation under cumulus clouds, but instead it presented a (very nice) sensitivity study to how the distributions of SSI change with cloud properties. I would like to ask the authors to rethink the exact purpose of this paper and to give the paper a clear and well-defined purpose. In my view, the best would be to focus the entire paper on the measurement strategy of SSI and create a dedicated study on SSI distributions and cloud properties, or to see how this information can be used to optimize measurement strategy.*

**Reply**: It is true that the paper contains two, somehow independent, parts. Our primary intention was to highlight the potential of the HOPE dataset to explore SSI variability, which had been so far mostly investigated using LES. This paper aims at showing the potential for further exploration of topics that definitely deserve dedicated papers, but would also require more care (accounting for aerosols in the simulations, using HOPE observations to extract actual cloud properties comparable to simulated ones etc.). The new title should help clarify this overarching objective, without detailing the secondary objectives. The introduction is also more explicit regarding the objectives and organization of the paper (at l.79):

"With the existing literature on SSI spatial variability in mind, the main objective of the present study is to investigate instantaneous SSI spatial distributions under cloudy conditions by combining the unique observations from the HOPE dataset with simulated SSI fields obtained by running 3D radiative transfer on LES simulated clouds. In line with previous studies addressing this question, we focus on cumulus clouds because they are responsible for the largest small-scale variability of SSI. These clouds, ubiquitous across a large fraction of the globe, also remain a challenge for weather and climate modelling, primarily because their small size means that they are generally parameterized, and their radiative impact as well. To identify situations from the HOPE dataset corresponding to golden cases of cumulus clouds, i.e., very homogeneous fields close to those simulated by ideal LES, we propose an original selection strategy. The comparison of these golden cases to simulations suggests that simulations are appropriate for studying SSI spatial distributions. Hence, building on this first general assessment of instantaneous SSI spatial distributions, we then tackle two independent questions. We first explore measurement strategies to

capture the SSI spatial distribution with a limited network of radiation sensors, which is addressed by combining the observations and simulations. We then investigate how cloud properties control SSI spatial distributions, which is carried out by perturbing the cloud properties in the simulation system and quantifying the impact on SSI distributions."

In the conclusions it is also suggested that both topics investigated would deserve dedicated studies:

l.453: "Although the measurement strategy investigation was limited to cumulus situations and did not consider the impact of aerosols, which could affect the results, this preliminary study was meant to demonstrate how simulations can be used to address this question. It certainly provides a valuable basis for further dedicated, more detailed studies, and paves the way for designing measurement strategies tailored for specific applications related to the high-resolution characterization of SSI."

l.461: "This sensitivity study is again somehow preliminary and would deserve a dedicated study, allowing to explore a variety of cumulus fields, not to say other cloud types. Importantly such future work should check whether the sensitivities highlighted by the simulations can be identified in the observations. For that purpose the HOPE dataset, gathering several remote sensing instruments, would be very relevant."

With these clarifications we hope the organization of the paper becomes clearer: In Section 3, we select some golden cases from the observations, to focus on ideal cumulus cloud conditions, and then compare (in Section 4), the pyranometers network observations of these golden cases to 3DRT simulations of a standard cumulus field simulated by an LES. This ensures that simulations can be used for further investigation. The last two sections focus on the two scientific questions: (1) measurement strategy for SSI variability, and (2) impact of cloud physical properties on the SSI distribution.

**Reviewer Point P 2.3** — *Table 2 contains many interesting insights in the impact of cloud properties on surface radiation, but only very few results are mentioned in the text. Please reflect deeper on this Table and the figures in the section, there is much more to learn from it.*

**Reply**: Table 2 is now much more detailed in the text, further developing the interpretation of the sensitivity analysis, and raising interesting questions for future work. The many modifications of Sect. 6 can be visualized in the track-changes version. We also clarify our limited intentions for the interpretation of the very rich Table 2 (l.384):

We do not aim at providing an exhaustive analysis of Table 2; instead, we focus on a few mechanisms and discuss how we understand them, this understanding resulting from the combination of many available sources of information (prior theoretical and bibliographical knowledge, tables and figures presented in this work).

**Reviewer Point P 2.4** — *Probably my biggest concern on the usefulness of the results of this paper is the absence of aerosols in the analysis and the radiation computations. As, for instance, Gristey has shown in his papers, aerosols largely change the probability distributions of SSI, and the direct/diffuse partitioning. If this paper is aiming for designing optimal measurement strategies, but at the same time performs computations on a situation that will not occur in reality, how applicable are the strategies then? I suggest the authors include at least some computations on aerosols to show the sensitivity.*

**Reply**: This is an important point also raised by the reviewer 1. Indeed, the simulations have been performed without any aerosols, and this is now clearly stated l.148:

"Importantly, the simulations are performed without aerosols, although they can significantly alter the SSI distribution (Gristey et al., 2022)."

The potential impact of aerosols is now explicitly mentioned in the discussion (l.465):

"As pointed out before, we did not consider aerosols and instead focused on the impact of clouds only. In reality, aerosols are ubiquitous, but their detailed representation in atmospheric and radiative models is challenging because their optical properties depend on their size and composition, but also on their hygroscopicity and ambient humidity. Several physical processes have also been identified that can explain the increase of aerosol optical depth in the vicinity of cumulus clouds (Eck et al., 2014). Besides this complexity, we did not have observational data to account properly for their effects throughout. Although aerosols were not accounted for in the simulations we believe that the qualitative results of the paper, along with the physical interpretations regarding the impact of cloud properties, would largely hold for real conditions. Yet, it is useful to discuss what impacts aerosols would have. According to Gristey et al. (2022), the presence of aerosols would typically shift the clear-sky mode to lower values due to increased absorption and the cloud-shadow mode to higher values due to extra radiation scattered laterally towards cloud shadows. Quantitatively, this impact could be as significant as those obtained from the sensitivity tests. In any case, accounting properly for aerosols would have required appropriate observations and a detailed optical module that are beyond the scope of the present paper. In view of making our understanding of the characteristics of surface flux distributions and their sensitivity to the overlaying atmosphere more exhaustive, future work should strive to include aerosols in simulations and perform additional sensitivity tests".

Besides these clarifications on the impact of aerosols, we have performed additional 3DRT simulations to study the impact of aerosols on the SSI pdf. The resulting simulations are the following:

- Control: without aerosol

- Case 0a: homogeneous concentration from the surface to cloud top such that total optical thickness $\approx 0.1$ (which is close to the similar study in Gristey et al. (2022))

- Case 0b: homogeneous concentration from the surface to cloud top such that optical thickness $\approx 0.15$

- Case 1: homogeneous concentration from the surface to cloud base, then 0 above, such that optical depth $\approx 0.1$

- Case 2: homogeneous concentration from the surface to cloud base such that optical thickness $\approx 0.1$ and homogeneous in the cloud layer such that optical thickness $\approx 0.05$. Total optical thickness $\approx 0.15$

- Case 3: heterogeneous concentration which is proportional to the concentration of water vapour such that optical thickness $\approx 0.1$ below the cloud base and optical thickness $\approx 0.05$ in the cloud layer

Figure 5 shows the results of these simulations, in terms of SSI distributions and fields. It confirms that the aerosol optical depth plays an important role in shifting the two peaks of the SSI distribution. The clear peak is shifted to the left and the shadow peak is shifted to the right, correlated to what is found in Gristey et al. (2022). The aerosols below the clouds (Case 1) have more impact than those within the clouds (Case 0a). In contrast, the spatial variability of aerosol seems to have less impact on SSI pdf. These results suggest that the overall shape of the SSI distribution is not altered by the presence of aerosols, so our interpretations of the Control case remain relevant for real conditions. More importantly,

as we do not pretend to derive a universal, operational strategy for pyranometer deployment, ignoring aerosols should not change the main ideas put forward in this paper.

Regarding the satisfactory match between the observed and clear-sky SSI pdfs, it probably comes from the fact that the conditions were relatively clean :

"They are both unimodal and symmetric, with approximately the same width and around the same mean value, suggesting that the impact of aerosols, which are not accounted for in the simulations, was rather limited for that particular day." (at l.271)

Note also that in dry conditions, the impact of aerosols is modest, compared to the very large AOD enhancement reported in the vicinity of clouds, and also simulated in Gristey et al. (2022) due to their hygroscopic growth. Hence the impact of aerosols is expected to be much larger in the presence of broken clouds due to what happens near cloud edges.

[Figure]

Figure 5: a preliminary study of aerosol's impact on the SSI pdf. (a) Vertical aerosol profile. (b) The SSI pdf with the presentation of aerosols. (c - i) The flux maps.

**Reviewer Point P 2.5**  —  *Why did the authors select cumulus clouds to do this study? Are they most relevant, or most appealing?*

**Reply**:  See our response to Point P2.1. Cumulus clouds are known for their impacts on short-term and small-scale variability of the SSI, not only because of the shadowing but also because of the cloud enhancement effect, where SSI can exceed clear-sky values due to reflections off cloud sides. The short-term and small-scale variability of SSI caused by cumulus clouds is critical for the stability and management of solar energy systems, such as photovoltaic (PV) farms. Understanding how cumulus

clouds affect SSI can help optimize the performance and reliability of these systems. Furthermore, Cumulus clouds present a challenge for weather and climate modelling due to their small size and complex interactions with solar radiation (they are responsible for strong 3D effects, currently uncaptured by plane parallel radiative transfer models). Hence studying these clouds can lead to improving their representation in atmospheric models. Despite the numerous field campaigns focused on cumulus clouds, there remains a lack of detailed observational data on their impact on SSI spatial distributions. This was the motivation for combining observational data with advanced modelling tools such as LES and 3DRT to gain insight into the mechanisms of how cumulus clouds impact SSI variability.

**Reviewer Point P 2.6** — *In my view, the selection procedure is unnecessarily complex. Why did the authors choose for such a detail level in selecting the case study?*

**Reply**: Indeed, the selection procedure is complex, but it is aimed at identifying golden cases of cumulus, with the primary intent to allow comparison with corresponding ideal LES. In addition, it proposes an objective way of identifying such situations, in comparison with cases selected in previous studies that were more subjective and mostly based on human observations. We believe that the selection procedure allows us to define an ideal cumulus field and will be useful for other applications. For information, we have also looked at the SSI distributions corresponding to periods that had been identified as cumulus clouds in Madhavan et al. (2017), and the distributions are quite different (see Fig. 6), suggesting that our approach is complementary for cumulus clouds identification. It is found that the periods identified by our selection procedure have lower shadow cover ratios, which are similar to our simulation cumulus case and favourable for comparing observation and simulation. Interestingly, periods identified in Madhavan et al. (2017) have larger shadow cover ratios, although the SSI distributions are very different from our selection periods, they still show the bimodal peaks.

**Reviewer Point P 2.7** — *The exact resolutions of the LES and ray tracer computations are a little unclear. It is stated that the RT is ran using 5x5m2 squares, but the LES is much coarser, so why is this high resolution used?*

**Reply**: The spatial resolutions of the 3DRT and LES are indeed different, the 3DRT resolution being much finer (5 m vs 25 m). Actually, even coarse clouds can result in fine features at the surface when considering rays passing near cloud edges, for instance, those contributing to cloud enhancement. Hence, to fully resolve the intensity and precise location of cloud enhancement, increasing the 3DRT resolution is crucial. To support this statement, Fig. 7 shows two simulated SSI maps and Fig. 8 shows the corresponding distribution with two 3DRT resolutions: 25 m (same as LES) and 5 m. Small differences can be noticed near cloud shadows' edges. A short explanation is added at l.139:

"Each pixel of each field is a $5 \times 5$ m$^2$ square. Note that a finer resolution than the LES is used to accurately simulate what happens near cloud shadows edges, where variations occur at smaller scales than the cloud resolution. Such a fine resolution allows to correctly simulate the rapid transition from the shadow to the clear-sky areas, and to capture the value of the maximum cloud enhancement, which is essential to reproduce the SSI distribution."

**Reviewer Point P 2.8** — *How are the photon paths distributed over the different spectral computations? There are 15000 per 5x5m2 pixel, but is this per quadrature point, per band, or in total?*

[Figure]

[Figure]

(a) 1 hour of cumulus days identified in Madhavan et al. (2017)

(b) Identified periods by our selection procedure

Figure 6: SSI distributions of days identified by our selection procedure and by Madhavan et al. (2017). The shadow cover ratio are included on the legends.

[Figure]

Figure 7: SSI map with different 3DRT resolution. Left: 25m. 3DRT uses the same resolution than it is used for LES. Middle: 5m. Right: Difference of the two SSI maps.

**Reply**: There are 15000 paths in total per 5x5m2 pixel. However, it is noted that we do not perform ray-tracing per quadrature point or per band. On the contrary, since we calculate the broadband flux, we sample a quadrature point within the spectrum integral for each photon path. This strategy does not

[Figure]

Figure 8: SSI pdf with different 3DRT resolution. blue: 25m. 3DRT uses the same resolution than it is used for LES. orange: 5m.

induce bias and fewer paths are needed compared to the band-by-band ray-tracing strategy (Villefranque et al., 2019). A short explanation is added at l.143:

"Each pixel corresponds to an SSI estimate, calculated as the mean flux over 15000 photon-path realizations, resulting in a statistical uncertainty of approximately 1%. Following the k-distribution model, a quadrature point within the spectrum integral is sampled for each photon path, following the method proposed by Villefranque et al. (2019). This strategy is proven to be unbiased and has good convergence performance."

**Other modifications**

- The acknowledgements are adjusted. A new financial declaration for this work is added and highlighted in blue.

- We also adjusted some English expressions throughout the document. These corrections aim to enhance the overall readability and clarity of the text.

**References**

Adhikary, S.K., Yilmaz, A.G., Muttil, N., 2015. Optimal design of rain gauge network in the middle yarra river catchment, australia. Hydrological processes 29, 2582–2599.

Eck, T., Holben, B., Reid, J., Arola, A., Ferrare, R., Hostetler, C., Crumeyrolle, S., Berkoff, T., Welton, E., Lolli, S., et al., 2014. Observations of rapid aerosol optical depth enhancements in the vicinity of polluted cumulus clouds. Atmospheric Chemistry and Physics 14, 11633–11656.

Gristey, J.J., Feingold, G., Schmidt, K.S., Chen, H., 2022. Influence of aerosol embedded in shallow cumulus cloud fields on the surface solar irradiance. Journal of Geophysical Research: Atmospheres 127, e2022JD036822.

Heinze, R., Dipankar, A., Henken, C.C., Moseley, C., Sourdeval, O., Trömel, S., Xie, X., Adamidis, P., Ament, F., Baars, H., et al., 2017. Large-eddy simulations over germany using icon: A comprehensive evaluation. Quarterly Journal of the Royal Meteorological Society 143, 69–100.

Madhavan, B.L., Deneke, H., Witthuhn, J., Macke, A., 2017. Multiresolution analysis of the spatiotemporal variability in global radiation observed by a dense network of 99 pyranometers. Atmospheric Chemistry and Physics 17, 3317–3338.

Papamichail, D.M., Metaxa, I.G., 1996. Geostatistical analysis of spatial variability of rainfall and optimal design of a rain gauge network. Water resources management 10, 107–127.

Riihimaki, L.D., Flynn, C., McComiskey, A., Lubin, D., Blanchard, Y., Chiu, J.C., Feingold, G., Feldman, D.R., Gristey, J.J., Herrera, C., et al., 2021. The shortwave spectral radiometer for atmospheric science: Capabilities and applications from the arm user facility. Bulletin of the American Meteorological Society 102, E539–E554.

Sedlar, J., Riihimaki, L.D., Lantz, K., Turner, D.D., 2021. Development of a random-forest cloud-regime classification model based on surface radiation and cloud products. Journal of Applied Meteorology and Climatology 60, 477–491.

Sentinel-Hub, . Sinergise-solutions. URL: `https://www.sentinel-hub.com`.

Villefranque, N., Fournier, R., Couvreux, F., Blanco, S., Cornet, C., Eymet, V., Forest, V., Tregan, J.M., 2019. A path-tracing monte carlo library for 3-d radiative transfer in highly resolved cloudy atmospheres. Journal of Advances in Modeling Earth Systems 11, 2449–2473.

Volkmann, T.H., Lyon, S.W., Gupta, H.V., Troch, P.A., 2010. Multicriteria design of rain gauge networks for flash flood prediction in semiarid catchments with complex terrain. Water resources research 46.

Xu, P., Wang, D., Singh, V.P., Wang, Y., Wu, J., Wang, L., Zou, X., Liu, J., Zou, Y., He, R., 2018. A kriging and entropy-based approach to raingauge network design. Environmental research 161, 61–75.